# *In silico* agent-based modeling approach to characterize multiple *in vitro* tuberculosis infection models

**Alexa Petrucciani**[1], **Alexis Hoerter**[1], **Leigh Kotze**[2], **Nelita Du Plessis**[2], **Elsje Pienaar**[1,3]*

**1** Weldon School of Biomedical Engineering, Purdue University, West Lafayette, IN, United States of America, **2** DSI-NRF Centre of Excellence for Biomedical Tuberculosis Research, South African Medical Research Council for Tuberculosis Research, Division of Molecular Biology and Human Genetics, Faculty of Medical and Health Sciences, Stellenbosch University, Cape Town, South Africa, **3** Regenstrief Center for Healthcare Engineering, Purdue University, West Lafayette, IN, United States of America

* epienaar@purdue.edu

**Data Availability Statement:** Data and code can be found at https://doi.org/10.5281/zenodo.7716557, https://doi.org/10.5281/zenodo.7716685, and

## Abstract

*In vitro* models of *Mycobacterium tuberculosis (Mtb)* infection are a valuable tool for examining host-pathogen interactions and screening drugs. With the development of more complex *in vitro* models, there is a need for tools to help analyze and integrate data from these models. To this end, we introduce an agent-based model (ABM) representation of the interactions between immune cells and bacteria in an *in vitro* setting. This *in silico* model was used to simulate both traditional and spheroid cell culture models by changing the movement rules and initial spatial layout of the cells in accordance with the respective *in vitro* models. The traditional and spheroid simulations were calibrated to published experimental data in a paired manner, by using the same parameters in both simulations. Within the calibrated simulations, heterogeneous outputs are seen for bacterial count and T cell infiltration into the macrophage core of the spheroid. The simulations also predict that equivalent numbers of activated macrophages do not necessarily result in similar bacterial reductions; that host immune responses can control bacterial growth in both spheroid structure dependent and independent manners; that STAT1 activation is the limiting step in macrophage activation in spheroids; and that drug screening and macrophage activation studies could have different outcomes depending on the *in vitro* culture used. Future model iterations will be guided by the limitations of the current model, specifically which parts of the output space were harder to reach. This ABM can be used to represent more *in vitro Mtb* infection models due to its flexible structure, thereby accelerating *in vitro* discoveries.

## Introduction

Tuberculosis (TB) continues to be a global public health crisis, responsible for 1.4 million deaths in 2021 alone [1]. TB is caused by *Mycobacterium tuberculosis* (*Mtb*). *Mtb* is introduced to its host upon inhalation of contaminated respiratory droplets, allowing direct entry into the

https://github.itap.purdue.edu/ElsjePienaarGroup/TB-in-vitro release v1.0.1

**Funding:** This work used the Extreme Science and Engineering Discovery Environment (XSEDE), which is supported by National Science Foundation grant number ACI1548562. Anvil at Purdue and Expanse at UCSD were used through allocation TG-MDE220002. This research was done using services provided by the OSG Consortium [63-66], which is supported by the National Science Foundation awards #2030508 and #1836650. The work of NdP is made possible through funding by the National Institute of Allergy and Infectious Diseases (NIAID), NIH, through contract #75N93019C0070. The work of AH is made possible in part by Grant TL1TR002531 (T. Hurley, PI) from the National Institutes of Health, National Center for Advancing Translational Sciences, Clinical and Translational Sciences Award.

**Competing interests:** The authors have declared that no competing interests exist.

lungs. Bacteria are deposited in the well-ventilated lower lobes of the lung, where alveolar macrophages phagocytose them [2]. *Mtb* is subsequently able to survive and replicate within the endosomes of these macrophages [3]. As the infection progresses, infected macrophages release chemokines and cytokines, which recruit other immune cells (e.g., monocytes, T cells, B cells, NK cells, dendritic cells, and neutrophils) to form a granuloma. A granuloma is generally comprised of a core of infected macrophages, surrounded by monocytes, epithelioid macrophages, foamy macrophages, neutrophils, multinucleated giant cells, and finally a lymphocytic cuff with an outer fibrous capsule [4]. The timing and spatial organization of key host-pathogen interactions within these granuloma structures, and how these interactions contribute to bacterial survival or elimination, remains incompletely understood. This is in part due to the complexity of the granuloma structure itself, which makes it difficult to understand, measure, and/or predict host-pathogen interactions and their impact on infection progression.

Many systems have been used to explore granulomas in TB; each having its own benefits and limitations. While much has been revealed about granuloma structure from humans studies, such studies are invasive or indirect and are often lacking in time points required to evaluate granuloma dynamics. Additionally, TB granulomas in humans can only really be studied at later stages when the infection has been established and diagnosed [5]. Animal studies such as non-human primate (NHP), rabbit, zebrafish, and mouse models are very useful and allow more control and direct observation of infection and granuloma formation than in humans. Mouse models benefit from the wide availability of commercial immunological reagents, genetic tools, and transgenic and knock-out strains, but most mouse strains struggle to recreate the structure of granulomas seen in humans [6,7]. Zebrafish have a natural pathogen pairing, can model active and persistent disease states, allow genetic perturbations of host and pathogen, and are conducive for imaging due to their transparent larval stage [8,9]. However, they only form non-necrotic, cellular granulomas and their natural pathogen *Mycobacterium marinum* is not identical to *Mtb* [8,9]. Rabbits and guinea pigs can form necrotic and non-necrotic mature granulomas [6,7]. These models have been limited in the past by the availability of immunological reagents, but recently more commercially available immunological reagents like antibodies against rabbit analytes have been developed [6,7,10,11]. NHP models most closely recreate human pathology, with heterogeneous clinical outcomes and granuloma structures [12,13]. But NHP models are expensive, time-intensive, and limited by the availability of animal facilities [6,7]. Overall, it is difficult to do certain genetic manipulations, collect data at many time points, and control the exact cellular and environmental makeup in these *in vivo* models.

Complementary to these *in vivo* models, more complex *in vitro* cellular cultures have been developed to both dissect biological mechanisms and test new therapies (reviewed in Elkington et al. [14]). *In vitro* models assemble mixtures of cells and bacteria in controlled ways in 2D (e.g. at the bottom of culture plates) or 3D (e.g. embedded in collagen matrices). Such complex *in vitro* cultures allow for mechanistic investigations that might not be feasible in the *in vivo* context. *In vitro* models can be particularly helpful because all cellular components can be controlled, and the models are tractable, meaning they can be more easily manipulated and observed than *in vivo* models. *In vitro* models are also cheaper and higher throughput than *in vivo* models.

Elkington et al. suggest certain criteria for an ideal *in vitro* TB model including the use of human cells and virulent *Mtb*; allowing incorporation of fibroblast, epithelial cells, and physiological extracellular matrix; being modular to allow many different biological questions to be answered; and, ideally, being 3-dimensional (3D) [14]. However, increasing complexity isn't necessary in all cases and can make models lower throughput and more expensive. Ideally, *in*

*vitro* models could be tailored to the biological question at hand, but could still be compared across experimental systems. *In vitro* models could then be optimized to include only the necessary components, allowing the maintenance of inexpensive, high-throughput models. Nonetheless, results from many disparate models can still be synthesized to form robust conclusions.

We recently developed an *in vitro* biomimetic 3D spheroid granuloma model [15]. Briefly, patient-derived alveolar macrophages are infected with BCG, and magnetic nanospheres are used to levitate the cells. Autologous adaptive immune cells isolated from peripheral blood mononuclear cells (PBMCs) were added at 48 hours into the 6-day culture. A traditional monolayer *in vitro* model is also cultured comprised of the same cells without structure or magnetic nanospheres. When comparing this granuloma model to the corresponding traditional monolayer culture, we found the spheroid model was better able to control bacteria. Differences in bacterial count between these models can be quantified and are due to the different model setups, but how the spatial aspects impact immune response is unclear. These two models provide a good test case to evaluate the possibility of translating between different *in vitro* models and identify the key mechanisms at work in the different models.

This data not only motivates a need to understand the mechanistic differences between these two models, but also highlights a need to more broadly look at the complexity and spatiality of *in vitro* models. As we move towards more complex *in vitro* models, organoids, complex cell mixtures, etc., it is important that we 1) understand and quantify the impact of the structural organization of the cultures, 2) develop tools that can analyze these more complex *in vitro* models, and 3) develop tools that enable us to compare and translate between *in vitro* models. Computational models are well-suited to address all of these tasks.

Mechanistic modeling has been applied to TB since 1962, and agent-based models (ABMs) in particular have been used in the context of TB since 2004 [16–18]. ABMs of granuloma formation in the non-human primate (NHP) lung have been iterated many times to look at the impacts of TNF-α [19–21], *Mtb* metabolism [22], macrophage (MΦ) polarization [23], and more [24–29]. The contributions of mechanistic modeling to the understanding of TB disease have been reviewed by Kirschner et al. 2017 and Pitcher et al. 2020 [17,30]. Notable contributions include the importance of IL-10 in infection outcome, something that was not predicted by mouse models, and the complex, dynamic balance of pro- and anti-inflammatory signals necessary to contain infection and limit pathology [17,31]. In this work, we apply these agent-based approaches to *in vitro* models rather than *in vivo* systems. This means that all components included in the *in vitro* model can be accounted for, the *in vitro* models can be more easily observed and perturbed, and we can use one simulation framework with different initializations to represent, and translate between, multiple *in vitro* models.

Computational models can aid the analysis of *in vitro* model data in two ways. First, if differences between *in vitro* models are small, or can be accounted for with computational model structure alone, the computational model can be calibrated to data from multiple *in vitro* models by only adjusting initializations to represent each *in vitro* model. Second, if a single computational model structure or parameter set cannot reproduce data from different *in vitro* models, this would indicate that the *in vitro* models are inherently different, and the computational models could generate hypotheses for these inherent differences.

In this work, we use one computational agent-based modeling framework to recreate the results from both the 3D spheroid and the corresponding traditional culture *in vitro* models [15]. We use these computational models to characterize the evolution of single spheroid and traditional cultures and quantify the heterogeneity of the host response within and between different spatial organizations. Finally, uncertainty analysis is used to identify similarities and differences between the spheroid and traditional *in vitro* models, and potential use cases are discussed.

## Methods

### Experimental methods

The data we use for calibration is derived from a biomimetic 3D spheroid model of a granuloma and the corresponding traditional culture (Fig 1) [15]. The experimental data that we use here were collected and published in a prior manuscript [15], and no new human samples or experimental measurements were collected for this manuscript. Briefly, HIV-negative patients with high suspicion of TB were recruited. Bronchoscopies were performed by qualified clinicians and nursing staff according to international guidelines [32] to obtain bronchoalveolar lavage fluid samples. Immediately after bronchoscopy, peripheral blood was collected by venipuncture into two 9mL sodium heparinized (NaHep) vacutainers. Alveolar macrophages were isolated from bronchoalveolar fluid, and PBMCs were isolated from peripheral blood using the Ficoll-Paque isolation method described previously [15]. Alveolar macrophages were cultured at a density of $4x10^5$ cells per well in a 24-well low-adherence culture plate and infected with *Mycobacterium bovis* Bacille Calmette-Guerin (BCG) at a multiplicity of infection (MOI) of 1 for 4 hours. Afterward, extracellular bacteria were removed by supplementing media with an antimycotic antibiotic (penicillin/streptomycin/amphotericin B) for 1 hour, followed by successive washes. The 3D spheroids were made by treating alveolar macrophages with biocompatible NanoShuttle (Greiner Bio-One) and levitating them using the magnetic levitating drive. After 48 hours, $6x10^5$ autologous CD3+ T cells are added per well. The traditional culture is made using the same cells and the same ratios but without NanoShuttle treatment and subsequent magnetic levitation.

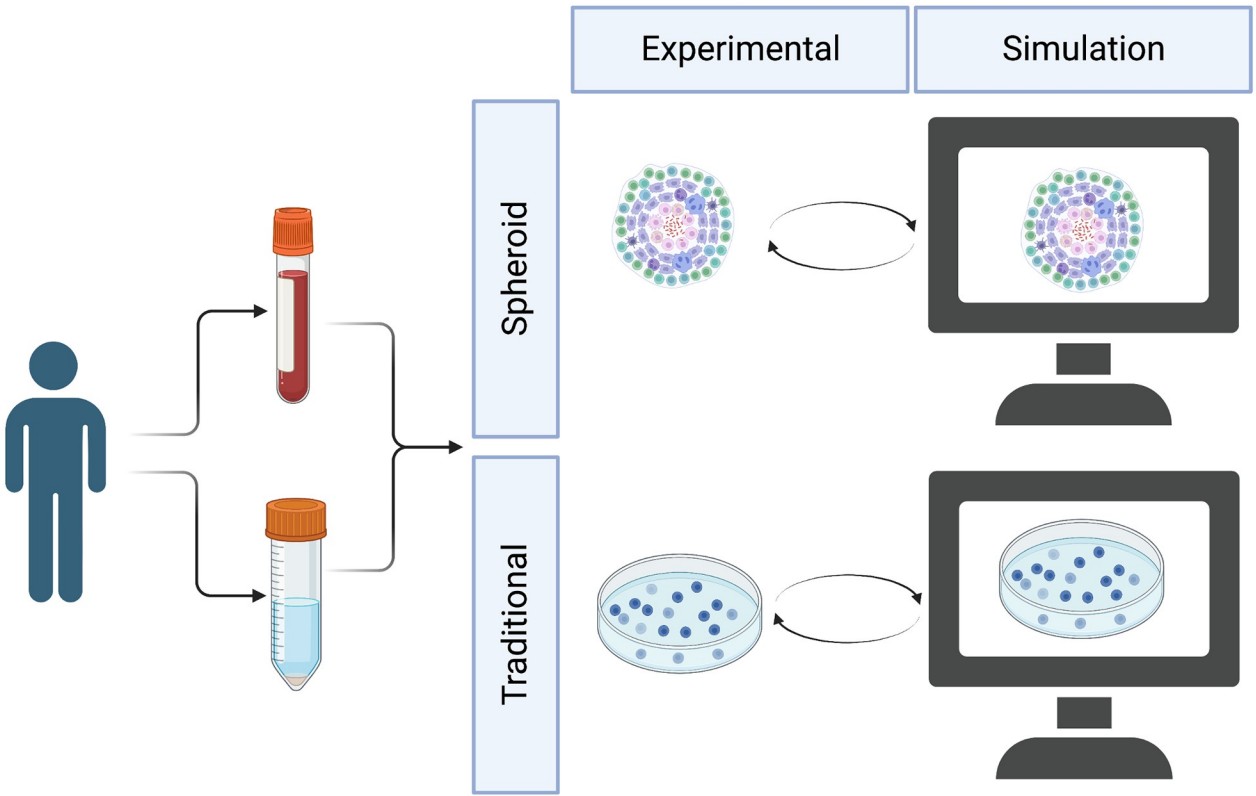

**Fig 1.** *In vitro* **and** *in silico* **spheroid and traditional models.** Peripheral blood and bronchoalveolar lavage fluid were collected from patients and recombined *in vitro* into spheroid and traditional experimental cultures. These cultures are modeled *in silico* with spheroid and traditional simulations. Created with BioRender.com.

Granuloma structures were mechanically disrupted by gentle pipetting after 6 days of culture. The day-6 timepoint was selected as the timepoint whereby the structure first demonstrated good stability and the time whereby we expect to detect measurable T cell immune responsiveness. The stability was determined by verifying that the macrophage plus T cell rim remains intact despite the removal of the magnet. Cell count and cell viability were determined using the trypan blue exclusion method after adherent cells were removed. Colony-forming unit (CFU) counts were determined by lysing mechanically disrupted cells and plating serial dilutions on Middlebrook 7H11 agar plates (BD Biosciences). After 21 days of growth, the colonies were manually counted. 3D spheroids were also fixed, embedded in tissue-freezing medium OCT (Tissue-Tek; USA), and cryosectioned. A section from the middle of the structure was stained with antibodies for CD3+ and CD206+ cells and imaged using a Carl Zeiss LSM 880 Airyscan with Fast Airyscan Module confocal microscope (Plan-Apochromat x63/ 1.40 oil DIC UV-VIS-IR M27 lens objective). The image of the traditional cell culture was acquired with light microscopy at 40x magnification. For full methods please reference Kotze et al. 2021.

## Model structure

Our model simulates the interactions between macrophages, CD4+ T cells, CD8+ T cells, bacteria, and two simplified cytokines within an *in vitro* environment. The simulation is constructed as a hybrid multiscale model with a cellular level agent-based model hybridized to a partial differential equation model of diffusion for the two cytokines (TNFα-like and IFNγ-like). These will be referred to as TNFα and IFNγ moving forward. The environment is composed of grid cubes that each represent a 20μm x 20μm x 20μm volume, which is the approximate size of our largest agent type, the macrophage [33]. The environment has two overlying grids, one single occupancy grid for immune cells and one multi occupancy for the smaller bacteria. The simulation has 4 types of agents: macrophages, CD4+ T cells, CD8+ T cells, and bacteria. Macrophages are subdivided into uninfected and infected classes. Agent behaviors are performed with a time step of 6 minutes, the approximate time for a monocyte to move 20 μm, or one grid cube [34–37]. The simulation is run for a total of 6 days, to reflect the duration of the *in vitro* experiments. An overview of agent behaviors is shown in Fig 2 and further detail is given in the text below and a flowchart of the simulation (S5 Fig). Parameters that are varied during calibration are given in Table 1 with initial ranges and references, and parameters kept constant are given in S1 Table in S1 File. These methods are in part drawn from work modeling NHP granulomas *in silico*, specifically GranSim and subsequent developments [17,18].

**Diffusing molecules.** There are 2 diffusing molecules included representing simplified TNF-α and IFN-γ. These are contributed to by the secreting agents and diffuse in the simulation space. TNF-α and IFN-γ were selected because they are known to be important in the immune response to TB, are well studied, and because they play major roles in included mechanisms: activation and chemotaxis [31]. We limited the number of cytokines to minimize the complexity of this first model iteration. Chemotaxis was modeled to be solely due to TNF-α. Anti-inflammatory cytokines like IL-10 were excluded due to the short time span of the experiment, based on evidence that IL-10 has minimal impact on bacterial burden or disease pathology early in the infection [52]. Diffusion is performed similarly to that in Weathered et al. using a 3D alternating-direction explicit numerical method [53]. As this method is unconditionally numerically stable, a larger *dt* than is predicted by the conditional stability criterion can be used while maintaining accuracy [54]. After finding *dt* suggested by the conditional stability criterion and the diffusion parameters a multiplier of 4 was incorporated into the

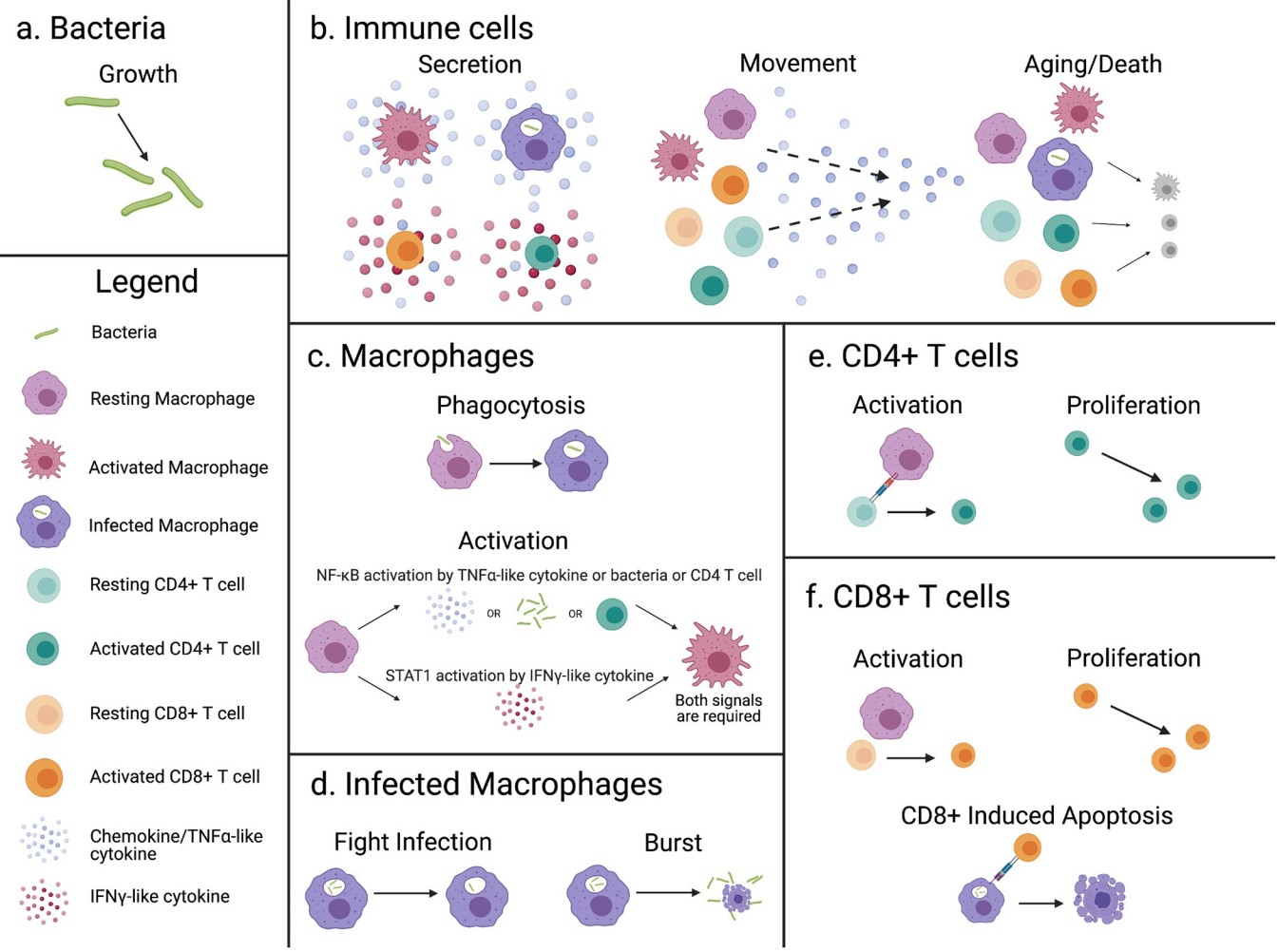

**Fig 2. An overview of rules for the simulated agents.** a) Bacteria grow and divide. b) Immune cells secrete cytokines dependent on activation or infectious state, move probabilistically up a TNF-α gradient, age, and die. c) Macrophages (MΦ) can phagocytose bacteria becoming infected. MΦ activation is represented by a nonsequential process that requires two signals: NF-κB and STAT1. NF-κB can be activated by TNF-α, bacteria, or direct contact with an activated CD4+ T cell. STAT1 is activated by IFN-γ secreted by activated T cells. d) Infected MΦ either fight infection by killing intracellular bacteria and returning to an uninfected state, or when a certain threshold of bacteria is reached will burst releasing intracellular bacteria into the environment. e) TB-Specific CD4+ T cells activate by interacting with a MΦ that has interacted with a bacterium. After activation, CD4+ T cells can proliferate. f) TB-Specific CD8+ T cells activate by interacting with a MΦ that has interacted with a bacterium and is STAT1 activated. After activation, CD8+ T cells can proliferate and kill infected macrophages along with the intracellular bacteria. Created with BioRender.com.

alternating-direction explicit method to reduce simulation time while maintaining accuracy, as recommended by Cilfone et al. [54]. This calculated PDE time step is then used to calculate the number of diffusion iterations needed per ABM time step. The PDE is run with a smaller time step than the ABM, ranging from 2 to 14 diffusion iterations per agent time step depending on the diffusion parameters and calculated stability criterion. IFN-γ and TNF-α are diffused separately with separate diffusion coefficients and decay rates. The rate of diffusion is slowed within granulomas by *granuloma fraction of diffusion*.

**Agents.** *Immune cells.* Macrophages, infected macrophages, CD4+ and CD8+ T cells are all classified as types of immune cells. These cell types were chosen to limit complexity, while including necessary components to form the core and cuff of the spheroid. Immune cells share common behaviors, including movement and aging. Movement is determined by gravity

**Table 1. Parameters that are varied during calibration.**

| Parameter | Initial Range | Units | Refs |
|---|---|---|---|
| **Bacteria** | | | |
| *Mtb internal doubling time* | 23,69 | Hours | [38] |
| *Mtb external doubling time* | 23,69 | Hours | [38] |
| **Macrophages** | | | |
| *activated macrophage proportion* | 0,0.1 | Per time step | e |
| *base killing probability* | 0.0001,0.02 | Per time step | e |
| *active killing probability* | 0.002,0.3 | Per time step | e |
| *base phagocytosis probability* | 0,1 | Per time step | f |
| *active phagocytosis probability* | 0,1 | Per time step | f |
| *phagocytosis threshold* | 8,12 | Intracellular bacteria | [21] |
| *cellular dysfunction threshold* | 8,12 | Intracellular bacteria | [21] |
| *NF-κB span* | 0.16,166 | Hours | [23] |
| *threshold for NF-κB activation TNF* | 40,500 | Molecules | e |
| *threshold for NF-κB activation bac* | 20,150 | External bacteria | [21] |
| *STAT1 span* | 0.16,166 | Hours | [23] |
| *threshold for STAT1 activation IFN* | 40,500 | Molecules | e |
| *activated macrophage TNF secretion* | 0,40 | Molecules/second | [39] |
| *infected macrophage TNF secretion* | 0,40 | Molecules/second | [39] |
| *macrophage population max lifespan* | 20,100 | Days | [21] |
| *macrophage population max activated lifespan* | 7,13 | Days | [21] |
| *base movement probability macro* | 0.5,1 | Per time step | [34–37] |
| *activated movement probability macro* | 0,0.5 | Per time step | e |
| **CD4+ T cells** | | | |
| *fraction CD4* | 0.5,0.65 | CD4+ T cells/ CD3+ T cells | [40,41] |
| *fraction TB-specific* | 0.0001,0.06 | TB-specific CD4+ T cells/ Total CD4+ T cells | [42,43] |
| *activated TB-specific CD4 fraction* | 0,0.1 | Initial activated TB-specific CD4 T cells/ Total TB-specific CD4 T cells | e |
| *activation probability CD4* | 0,1 | Per time step | f |
| *deactivation probability CD4* | 0,1 | Per time step | f |
| *activated CD4 TNF secretion* | 0,40 | Molecules/second | [39] |
| *activated CD4 IFN secretion* | 0,40 | Molecules/second | [39] |
| *CD4 population doubling time* | 6,16 | Hours | [44,45] |
| *maximum CD4 generations* | 3,10 | Generations | [44,46,47] |
| *CD4 population max lifespan*<br>*CD8 population max lifespan* | 34,340 | Days | [48–50] |
| *CD4 population activated lifespan*<br>*CD8 population max activated lifespan* | 2.5,4 | Days | [21,44] |
| *base movement probability CD4*<br>*base movement probability CD8* | 0,1 | Per time step | f |
| *activated movement probability CD4*<br>*activated movement probability CD8* | 0,1 | Per time step | f |
| **CD8+ T cells** | | | |
| *fraction CD8* | 0.3,0.35 | CD8+ T cells/ CD3+ T cells | [41] |
| *TB-specific CD8 fraction* | 0.0001,0.06 | TB-specific CD8+ T cells/ Total CD8+ T cells | [42,43] |
| *activated TB-specific CD8 fraction* | 0,0.1 | Initial activated TB-specific CD8 T cells/ Total TB-specific CD8 T cells | e |
| *activation probability CD8* | 0,1 | Per time step | f |
| *deactivation probability CD8* | 0,1 | Per time step | f |
| *activated CD8 TNF secretion* | 0,40 | Molecules/second | [39] |
| *activated CD8 IFN secretion* | 0,40 | Molecules/second | [39] |

*(Continued)*

**Table 1.** (Continued)

| Parameter | Initial Range | Units | Refs |
|---|---|---|---|
| *CD8 population doubling time* | 3,13 | Hours | [45] |
| *maximum CD8 generations* | 7,20 | Generations | [46,47,51] |
| *kill probability CD8* | 0.012,0.12 | Per time step | [21] |
| **Diffusion** | | | |
| *threshold for immune cell movement TNF* | 1,500 | Molecules | e |
| *diffusion coefficient TNF* | 0.1,1 | 10^-7 cm^2/s | [23] |
| *degradation rate per second TNF* | 0.96,10 | 10^-4 1/s | e |
| *diffusion coefficient IFN* | 0.1,1 | 10^-7 cm^2/s | [23] |
| *degradation rate per second IFN* | 0.96,10 | 10^-4 1/s | e |
| *granuloma fraction of diffusion* | 0,1 | - | f |
| *sphere efficiency* | 0.65,0.9 | - | e |

Initial ranges are either determined by literature, estimated through preliminary simulations (e), or broadened to the full mathematically possible range (f). Initial ranges were all sampled with uniform distributions. All parameters are continuous, but *phagocytosis threshold*, *cellular dysfunction threshold*, and *maximum CD{4,8} generations* behave discretely as they are only used in comparison to integer values. One time step is equivalent to a 6-minute interval. Note that *CD{4,8} population max lifespan*, *CD{4,8} population activated lifespan*, *base movement probability CD{4,8}*, and *activated movement probability CD{4,8}* are shared for CD4+ and CD8+ T cells and are therefore only counted as 4 parameters. The set of calibrated parameter sets can be found in the supplemental data with distributions shown in S1 Fig.

limited or 3D rules. Cells moving in 3D can move in any direction. With gravity limited rules, cells will fall in the z dimension if no immune cell is below them and can only move up in the z direction if on top of another immune cell. Given these movement rules, the cells will chemotax probabilistically toward the highest concentration of TNF-α when the summed TNF-α in the Moore neighborhood, the 26 grid cubes that directly border the given square, is above *threshold for immune cell movement TNF*. This chemotaxis algorithm is based on that in Weathered et al. [53]. Immune cells also age according to individualized lifespans. A resting lifespan and activated lifespan are selected for each cell from a *population lifespan* * (1+/- *lifespan variance*). These lifespans are then converted to aging rates, which change according to the activation status of the cell. The resting aging rate is 1 hour aged per hour, while the activated aging rate is calculated as resting lifespan divided by activated lifespan. At initialization, a cell will be given a random starting age from zero to the resting lifespan. Then a cell's current age gets incremented by the aging rate each time step. When a cell reaches its maximum age, it will die and be removed from the simulation.

*Macrophages*. Beyond the immune cell rules described above, macrophages will attempt to phagocytose and activate every time step. Each macrophage attempts to phagocytose by picking a bacterium in its Moore neighborhood at random. If this bacterium is extracellular, it will be phagocytosed with a phagocytosis probability dependent on the activation state (*base phagocytosis probability*, *active phagocytosis probability*). Successful phagocytosis turns a macrophage into an infected macrophage. Macrophages that have phagocytosed bacteria also get classified as having interacted with bacteria, meaning antigenic peptides can be displayed on the cell surface. Each macrophage also checks for activation. Activation is represented by a simplified two-step signaling process, requiring STAT1 and NF-κB activation [55]. STAT1 is activated if local IFN-γ is greater than *threshold for STAT1 activation IFN*. NF-κB can be activated in 3 ways: TNF-α greater than *threshold for NF-κB activation TNF*, nearby extracellular bacteria greater than *threshold for NF-κB activation bac*, or direct interaction with an activated CD4+ T cell. These represent TNF-α interaction with TNFR, activation of TLR, and CD40-CD40L interactions, respectively [56]. NF-κB and STAT1 activations last for set

durations after the signal was initially received (*NF-κB span* and *STAT1 span*). After the macrophage-specific length of activated time, the pathway will deactivate and be checked again immediately, to allow longer activation if the activation signals persist. If both pathways are activated at the same time, then the macrophage becomes fully activated. Activation changes a macrophage's movement probability, phagocytosis probability, and aging rate. Activated macrophages also secrete TNF-α at a rate of *activated macrophage TNF secretion* molecules per second.

*Infected macrophages*. Infected macrophages can fight the infection at each time step. Each infected macrophage corresponds to a list of one or more intracellular bacteria that have been phagocytosed by the macrophage and exist in the same location on the corresponding bacterial grid. An intracellular bacterium is selected randomly and will be killed with a probability that is dependent on the macrophage's activation state (*base killing probability*, *active killing probability*). If all the bacteria within an infected macrophage are killed, then the infected macrophage reverts to a healthy macrophage. Infected macrophages can be activated through the same pathways as healthy macrophages. When fully activated, the phagocytosis and killing probabilities change to values for activated macrophages. Infected macrophages secrete TNF-α when activated, but also constitutively secrete TNF-α at a baseline level of *infected macrophage TNF secretion* molecules per second when not activated. Infected macrophages don't move but can continue to phagocytose bacteria if the number of intracellular bacteria is below *phagocytosis threshold*. Once the number of intracellular bacteria is above *cellular dysfunction threshold* the macrophage is considered chronically infected [18]. Chronically infected macrophages can no longer be fully activated or kill intracellular bacteria. If the number of bacteria within an infected macrophage reaches a bursting threshold the macrophage will burst and release the intracellular bacteria into the environment. A burst limit was randomly selected for each infected macrophage from a uniform distribution from 20 to 40 intracellular bacteria [57]. When a macrophage dies of old age the bacteria are similarly released into the environment.

*CD4+ T cells*. CD4+ cells can be TB-specific or non-TB-specific. TB-specific CD4+ T cells can also become activated. Activation of TB-specific CD4+ T cells occurs with a probability of *activation probability CD4* if a random macrophage in its Moore neighborhood has interacted with bacteria. This is equivalent to antigen presentation on MHC II [56]. Activation increases movement probability and aging rate. Activated CD4+ T cells secrete both TNF-α and IFN-γ at *activated CD4 TNF secretion* and *activated CD4 IFN secretion* molecules per second, respectively [39]. Active CD4+ T cells can also divide with a doubling time of *CD4 population doubling time* until the maximum number of generations (*maximum CD4 generations*) is reached. If there is no space available when a cell divides, it is recorded as a failed attempt. An average of less than 5% of attempted divisions fail in calibrated runs. This could be thought of as nutritional inhibition that prevents the cells from growing. Deactivation occurs with a given probability *deactivation probability CD4* per time step.

*CD8+ T cells*. Just like CD4+ T cells, CD8+ T cells can be subdivided into TB-specific and non-TB-specific. TB-specific CD8+ T cells can be activated. If a randomly selected macrophage within the T cell's Moore neighborhood is STAT1 activated and has interacted with bacteria, then the T cell will probabilistically activate (*activation probability CD8*). STAT1 activation is a proxy for interaction between CD4+ T cell and macrophage which increases expression of molecules on the surface of the APC(B7 and 4-1BBL) that provide co-stimulation to naïve CD8+ T cells [56,58]. If activated, a CD8+ T cell will secrete both TNF-α (*activated CD8 TNF secretion*) and IFN-γ (*activated CD8 IFN secretion*). Activation also increases movement probability and aging rate. Activated CD8+ T cells will also divide with a doubling time of *CD8 population doubling time* until the maximum generation (*maximum CD8 generations*) is reached. Activated CD8+ T cells can kill infected macrophages (equivalent to cells

presenting peptides in MHC I). A random infected macrophage is selected for the Moore neighborhood, and the infected macrophage and all intracellular bacteria are killed with a probability *kill probability CD8*. CD8+ T cells deactivate probabilistically (*deactivation probability CD8*).

*Bacteria*. Bacteria grow and divide. Bacteria have biomass that gets added to every time step. The rate of growth depends on whether they are intracellular or extracellular. Growth rate is calculated from doubling time (*Mtb internal doubling time*, *Mtb external doubling time*) and includes some individual variance from the population mean. If the biomass threshold of 2 is reached, then the bacteria divide into two with the biomass distributed among them unevenly [59]. Simulated bacteria represent BCG, as BCG was used in the *in vitro* models. Behaviors/parameters are drawn from both BCG and TB literature.

**Initial conditions.** The differences between the spheroid and traditional simulations include the movement rules and the initial spatial distribution of cells. Our initial conditions reflect those used in the *in vitro* models [15].

*Spheroid*. In the experimental protocols, 400,000 macrophages are infected with MOI 1 and then levitated [15]. While the magnetic beads aren't explicitly included in the model, they impact the movement rules of the cells by allowing them to move in 3D as described in the immune cells section. After two days, 600,000 CD3+ cells are added in a dropwise manner directly to the spheroid. Due to computational limitations associated with the 3D simulation of a full-sized spheroid, we simulate a spheroid of 1/10$^{th}$ the size. Due to this downscaling, calibrated parameters are assumed to be effective parameters. Scaling analysis of representative parameter sets shows that larger spheroid simulations (up to one-half of experimental structures) show similar overall behavior to our simulations at 1/10$^{th}$ size. (S2 and S3 Figs) We generate a sphere of 40,000 mixed healthy and infected macrophages. Given the experimental MOI of 1, we use a Poisson distribution to estimate the percentage of cells with various numbers of phagocytosed bacteria [60], giving the fraction of macrophages that have phagocytosed *n* bacteria as $\frac{MOI^n e^{-MOI}}{n!}$. Macrophages with zero to six intracellular bacteria are initialized, giving 39,997 initial bacteria. This sphere is centered on an 80x80x80 grid representing 1.6 mm x 1.6 mm x 1.6 mm volume. The radius of the initialized sphere is calculated as $\sqrt[3]{\frac{40,000*3}{4 \pi \, sphere \, efficiency}}$, with the initial density of the cells determined by *sphere efficiency*. *Sphere efficiency* is the fraction of grid cubes that are initialized to contain cells. It has a maximum value of 1 where every grid cube has a cell resulting in a smaller, more densely packed sphere. Lower values have empty grid cubes within the structure, resulting in a larger, less dense structure. At day 2, 60,000 CD4 + and CD8+ T cells are added in a cuff around the macrophages. Proportions of CD4+ T cells (*fraction CD4*), CD8+ T cells (*fraction CD8*), and TB-specific T cells (*fraction TB-specific, TB-specific CD8 fraction*) are estimated from literature [40–43]. Subsets of the immune cells are allowed to be preactivated (*activated macrophage proportion*, *activated TB-specific CD4 fraction*, *activated TB-specific CD8 fraction*) as the alveolar macrophages and PBMCs were taken from patients with active TB. Activated TB-specific T cells are given a random starting generation and starting point in the division cycle as the process of proliferation could have already started.

*Traditional culture*. The experimental conditions are the same as the spheroid without the inclusion of the magnetic levitation beads. The exclusion of the magnetic beads is represented in the simulation by gravity limited movement rules as described in the immune cells section. As with the spheroid, the traditional simulation is 1/10$^{th}$ the size of the experiment. This is simulated by adding 40,000 infected and uninfected macrophages distributed evenly through the environment. After these macrophages are added they fall to the bottom of the plate due to the gravity-limited movement. Since the cells would all be at the bottom of the plate, the

dimensions were adjusted to 216 x 216 x 11, or 4.32mm x 4.32mm x 0.22 mm. The ratio of cells to the surface area of the plate is initialized to be the same in the simulation as in the *in vitro* model. Additionally, the volume of simulation, and therefore initial cellular density, is minimally different between the spheroid and traditional models. The percentage of cells with various numbers of phagocytosed bacteria is calculated in the same manner as the spheroid model. At day 2, 60,000 CD4+ and CD8+ T cells are distributed evenly throughout the environment before falling. All parameters are kept the same for the spheroid and traditional simulation, except the parameters that dictate initial structure and gravity-limited movement (*case number* and *is gravity* in S1 Table of S1 File).

**Simulation and parameter sampling.** This model is built using Repast Simphony 2.8, an open-source software used to build ABMs in Java [61]. The Repast model code is located at https://github.itap.purdue.edu/ElsjePienaarGroup/TB-in-vitro release v1.0.1. Simulations were run on the Purdue Brown Cluster, XSEDE, and OSG resources [62–66]. Python and MATLAB were used for data analysis and visualization. Some parameters are varied between cells of the same type within a population to represent biological heterogeneity. Variances are specified in S1 Table of S1 File, and cellular parameters are sampled uniformly within [parameter*(1-variance), parameter*(1+variance)]. Probabilities were not sampled between cells because heterogeneity in these cases is established through the stochasticity associated with the comparison of randomly generated values to these probability parameters. Other parameters that were held constant between all cells in a population include secretion rates, maximum generations (i.e., the number of times a cell can proliferate), and thresholds for activation, movement, and cellular dysfunction.

## Calibration

Calibration is performed by doing an initial parameter sweep and then iterating around specific parameter sets. These iterations are used to find a variety of parameter sets that fit into the experimental data range while iterating to reach parts of the output space that were not captured in the previous sample. Experimental data ranges used for calibration include:

- Spheroid bacterial fold change from 4 hours post-infection to day 6

- Traditional bacterial fold change from 4 hours post-infection to day 6

- Spheroid cell viability at day 6

- Traditional cell viability at day 6

- Spheroid cell count at day 6

- Traditional cell count at day 6

A total of 50 parameters are varied in the model (Table 1). Initial ranges are determined from relevant literature (*in silico*, *in vivo*, *in vitro*) or left broad if no estimates were identified in the literature. A schematic of the calibration procedure for 2 parameters and 2 model outputs is provided in Fig 3. Latin hypercube sampling (LHS) was used to sample 1,000 parameter sets from initial ranges with a centered design (Table 1). These parameter sets are run in both the traditional and spheroid simulation with 7 replicates as a broad initial sweep. Top runs are defined as those with the highest traditional CFU, as this part of the output space had few runs in the initial sweep. The top five runs that met the bacterial fold changes for traditional and spheroid are iterated. Iterations are performed by narrowing the parameter range to 20% of the initial range centered around the initial point (each of the top five runs). One hundred samples in this new range are generated using LHS and are run in triplicate. The number of

## a. Initial Sweep

## b. For Each Enrichment Point

**Fig 3. Calibration method used for targeted exploration of the output space.** This is a simplified example with only two parameters and output dimensions to visualize steps. A) An initial sweep was performed with the entire parameter space. Traditional and spheroid simulations were run with all parameter sets (blue shapes). The output space has experimentally determined ranges that are used for calibration represented by grey regions, the desired output space is the overlap of these regions (green region). We used this method to enrich desired parts of the output space in a directed manner. In this case, higher output 2 is desired (blue arrow). Thus, points that fall in the overlap of the experimental regions (green region) with the highest output 2 are selected to iterate. (⬠) These are called enrichment points. B) For each of these enrichment points, a new region is defined around the initial parameter set by narrowing the parameter range to 20% of its initial sampled range as noted by the smaller grey box. This range is sampled using LHS and run through the simulations. The point with the highest output 2 that falls in the experimental region is selected as a new enrichment point (▼), and this process is repeated until less than a 10% increase in output 2 is seen. The region is then further narrowed to 10% of the initial range, and iterations are performed as above. The calibrated set is then defined as all parameter sets that meet the six calibration criteria.

replicates and runs are reduced due to computational costs. Runs that passed all 6 criteria (bacterial fold changes, cell viability, and cell count at day 6 for traditional and spheroid cultures) are iterated until there was less than a 10% increase in traditional culture CFU. The iterating range is then narrowed to 10% of the initial range, and iterated until again there is a less than 10% increase in traditional culture CFU. The calibrated set is generated by selecting runs that fit all 6 criteria from all the simulations, giving us a collection of parameter sets with all 50 parameters varied. Thus, our approach allows us to enrich areas that fall within experimental ranges while directing the traditional CFU higher in order to fill out the whole experimental range.

## Uncertainty analysis

LHS and partial rank correlation coefficients (LHS-PRCC) are used to perform an uncertainty analysis [67]. LHS-PRCC has been used with similar simulations to characterize monotonic relationships between inputs and outputs [67]. One thousand samples are selected from the initial range using LHS and run with 7 replicates. These replicates are averaged before PRCCs are calculated at day 2 before the T cells are added and day 6. A significance level of 0.01 is used with a Bonferroni correction for the number of tests run. The relationship between the 50 varied parameters and 9 outputs of interest (total *Mtb* count, *Mtb* killed by activated macrophage count, *Mtb* killed by resting macrophage count, *Mtb* killed by CD8+ T cell count, activated CD4+ T cell count, total activated CD8+ T cells, activated macrophage count, total STAT1 macrophage count, total NF-κB macrophage count) are analyzed.

## Matching unpaired runs

To be able to explore simulations that reach the upper ranges of traditional CFU counts, something that was not achievable using paired simulations, we also analyze matched simulations. Unpaired spheroid and traditional simulations are matched by selecting runs with similar (but not identical) initial condition parameters: *fraction CD8*, *fraction CD4*, *fraction TB-specific*, and *TB-specific CD8 fraction*. To identify matched simulations, the spheroid runs are looped through for each traditional run, and a cost function is calculated. This function sums squared errors divided by the maximum value for these 4 controlled parameters (*fraction CD8*, *fraction CD4*, *fraction TB-specific*, and *TB-specific CD8 fraction*).

$$\sum_{i \in A} \left( \frac{p_i - p_{imatched}}{\text{upperBound}(p_i)} \right)^2$$

$$A = \{fraction\ CD8, fraction\ CD4, fraction\ TB - specific, TB - specific\ CD8\ fraction\}$$

Where $p_i$ is the parameter value in the spheroid simulation, $p_{imatched}$ is the potential matching parameter value in the traditional simulation, and the upper bound is the upper bound of the initial range as given in Table 1. The spheroid simulation with the lowest cost is selected to be matched to the unpaired traditional run.

# Results

## Results from multiple *in vitro* cultures are reproduced with one in silico framework and shared parameters

We first test if the multiscale model can recreate the experimental data for bacterial fold change, cell count, and cell viability at day 6 by only varying initial spatial layout and movement rules between the traditional and spheroid simulations. Using the calibration method described above, parameter sets are identified whose outputs fit the criteria for both spheroid and traditional *in vitro* data. (Fig 4A–4C) Our simulations produce CFU fold change outputs that span most of the experimental range, except for the highest experimentally measured CFUs in the traditional cultures. Together this indicates that experimental data from spheroid and traditional *in vitro* cultures can be reproduced using the same sets of parameters (S1 Fig) and the same model structure.

After calibrating to both spheroid and traditional cultures, representative calibrated runs are visualized to compare with experimental images as a qualitative validation. Simulated spheroids (Fig 4G and 4H) qualitatively match experimental microscopy (Fig 4F), having a layered structure with macrophages on the inside and T cells in a cuff around the edge. The

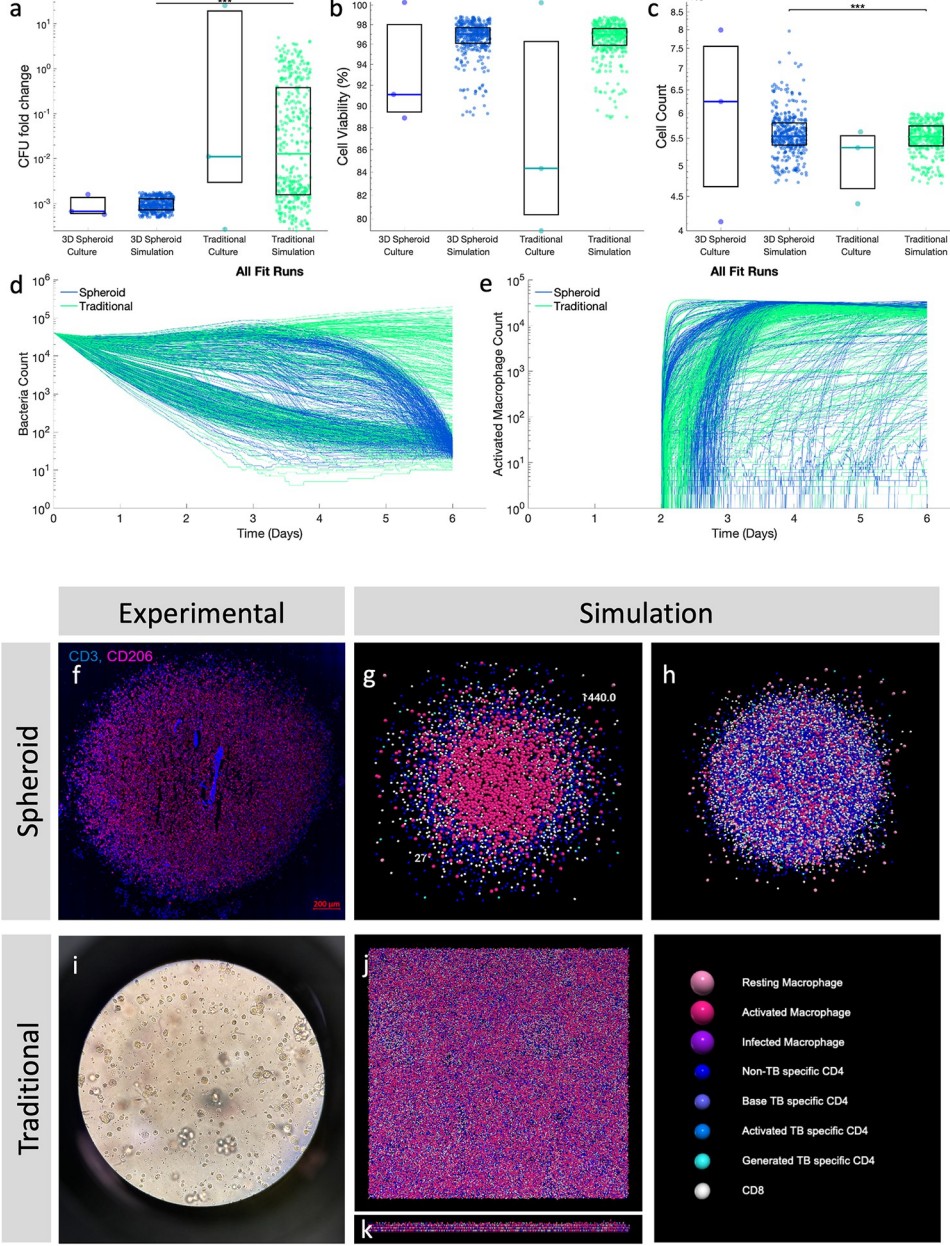

**Fig 4. Paired simulations are calibrated to data from *in vitro* cultures.** Spheroid and traditional simulations are run with the same parameters, only varying the initial spatial layout of cells and the movement rules between the two simulation types. Comparison of experimental data (n = 3) to calibrated simulation data (n = 398) for a) CFU fold change from 4 hours post-infection to 6 days, b) cell viability at day 6, and c) cell count at day 6. d) Bacterial count, and e) activated macrophage count dynamics for calibrated spheroid and traditional simulations over the 6-day time course show heterogeneous behaviors. Representative spheroid and traditional simulations are visualized at day 6 for comparison to *in vitro* images. f) A slice of the *in vitro* spheroid culture at day 6. (Adapted with permission from Kotze et al. 2021) g) A slice through the center of a spheroid simulation at simulated day 6, or the 1440th six-minute time step. The number in the top right is the Repast time step, and the number in the bottom left is the simulation number. h) Full spheroid simulation. i) A brightfield image of the *in vitro* traditional culture on (day 6). j) Traditional simulation viewed top down. k) Traditional simulation viewed from the side. *** p≤1e-3.

spheroid has a dense center with some cells less densely around the outside. The layered structure of the spheroid can be contrasted with the more well-mixed traditional simulation (Fig 4J and 4K) and experiments (Fig 4I). In the traditional simulation, cells are localized at the bottom of the simulation space, due to the gravity-limiting spatial rules. In summary, quantitative calibration and qualitative validation indicate that our simulation-predicted temporal outputs and spatial organization align well with experimental data.

Beyond recreating existing experimental data, our computational model predicts that there is a heterogeneous group of bacterial time courses that are consistent with the experimental data at day 6. (Fig 4D) Bacterial dynamics range from traditional simulations that show minimal bacterial reduction following the addition of T cells at day 2, to traditional and spheroid simulations that show transient bacterial reduction after T cell addition, and traditional and spheroid simulations that show sustained bacterial reduction throughout the entire simulation. These diverse bacterial dynamics are associated with similarly diverse activated macrophage dynamics (Fig 4E). However, the activated macrophage dynamics reveal that, in most simulations, the majority of macrophages become activated. This suggests that more complex spatial mechanisms are driving the diversity in bacterial dynamics. Our predicted macrophage activation (Fig 4E) could be compared with M1 activation markers *in vitro* at day 2.5 to differentiate between the spheroid simulations with different predicted timings of macrophage activation.

Taken together, these results indicate that our computational framework is able to reproduce both aggregate and spatial data from *in vitro* spheroid and traditional cultures. Additionally, our simulations indicate that activation of a large number of macrophages is not always sufficient to reduce bacterial numbers in these *in vitro* cultures.

## Spatial evolution of spheroid simulations indicates that macrophage activation timing is limited by T cell interaction and macrophage location within spheroids

To examine when and where macrophages become activated within spheroids we next analyzed, both visually and quantitatively, the spatial evolution of macrophage activation in a representative spheroid simulation. An initial sphere of mixed infected and uninfected macrophages is present at day 1 with a cuff of T cells being added at day 2 (Fig 5A). Macrophage activation starts at the interface of the macrophages and T cells and moves toward the center as time progresses. This activation coincides with T cell infiltration into the macrophage core. We calculate the radial density of cell subpopulations to quantitatively investigate these trends (Fig 5B–5E). The simulation starts with uniformly distributed macrophages before a cuff of uniformly distributed CD4+ and CD8+ T cells is added (Fig 5B). As time progresses the T cells spread out and begin to infiltrate the macrophage core. In this representative simulation there is more CD8+ T cell activation leading to more CD8+ T cells infiltration. Activated T cells make up the minority of the T cell populations and are more localized towards the center of the granuloma (Fig 5C). This makes sense as interacting with a macrophage presenting antigenic peptides is required for T cell activation, and bacteria and macrophages that have interacted with bacteria are localized to the core. These activated T cells provide one of the signals required for macrophage activation, STAT1 via IFN-γ. Indeed, when the distribution of the IFN-γ-driven STAT1 activation signal is overlaid with the other required signal for macrophage activation, NF-κB, results indicate that macrophage activation propagates from the outside in (Fig 5D). The distribution of fully activated macrophages (Fig 5E) closely follows the STAT1 signal, further indicating that in this model T cells (and not NF-κB activation) are the limiting step of macrophage activation. This NF-κB activation is solely due to TNF-α secretion

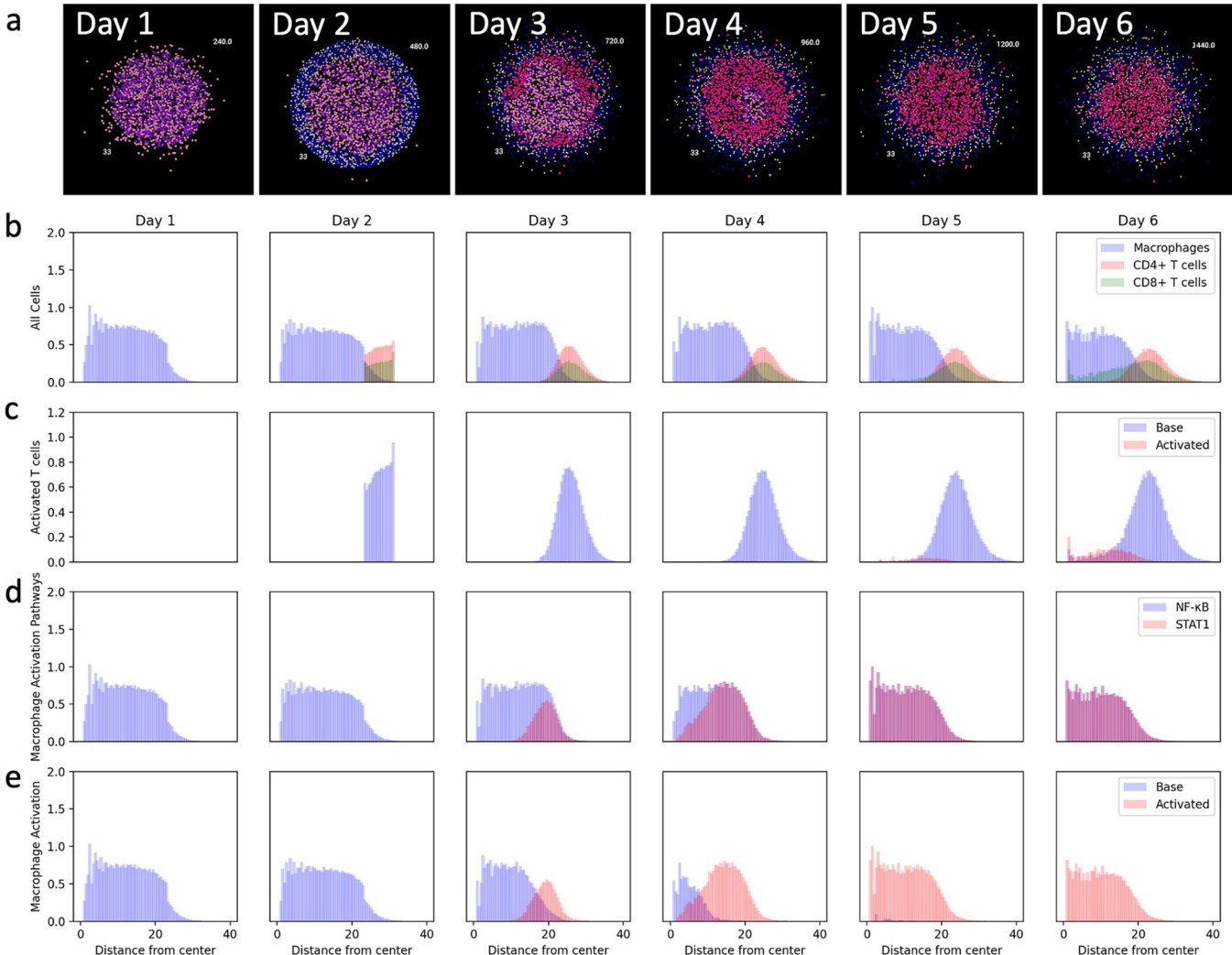

**Fig 5. Spatial distribution of cells in a granuloma over time.** a) The spatial development of a single granuloma over 6 days. The radial distribution of b) macrophages, CD4+ T cells, and CD8+ T cells; c) base and activated T cells; d) NF-κB and STAT1 activated macrophages; e) base and activated macrophages. The y-axes represent the radial density of cells. Radial density graphs were generated by calculating the distances of the cells to the center of the spheroid, generating a histogram for the cells of interest by dividing them into preset bins, and then normalizing by the total volume in each bin which corresponds to the volume of a spherical shell. S4 Fig shows the same graph averaged across all simulations.

from the infected macrophages, with 100% of NF-κB activation being due to TNF-α during the first two days of simulation.

Taken together, this illustrates how our models are used to predict and quantify key host-pathogen interactions in space and time in a single granuloma. Our simulations predict that IFN-γ-driven STAT1 activation is the limiting factor in macrophage activation in spheroids, that macrophage activation occurs at later time points for macrophages at the center of the spheroids since it depends on T cell infiltration, and that relatively few activated T cells are responsible for macrophage activation within the infected spheroid core. These distributions could be tested by staining and imaging slices of spheroids over time, for example the activation markers for macrophages at days 3 and 4.

## A distribution of outcomes, including T cell infiltration, is seen among spheroid simulations

Given the predicted importance of T cell infiltration for macrophage activation in the spheroid simulations, we investigate T cell infiltration in our diverse collection of spheroid granulomas. Spheroid simulations show variation in T cell infiltration between parameter sets (Fig 6A and 6B). Calibrated runs vary from no infiltration (Fig 6A) to an almost homogeneous mixing of macrophages and T cells (Fig 6B). To evaluate the heterogeneity across all of our simulations, the mean and standard deviation of radial density is calculated for macrophages and T cells for each simulation. This gives a mean position when correcting for the uneven volumes of the radial spheres. As suggested by the representative examples in Fig 6A and 6B these position metrics for T cells and macrophages range from having nearly complete overlap to almost complete separation (Fig 6C). Little infiltration was seen in the *in vitro* model at day 6 [15], which aligns with some but not all of our simulated runs. Either the small sample size of the experimental study doesn't account for full heterogeneity, or this information can be used to further narrow the parameter space in future model iterations. Cell segmentation on a larger number of stained *in vitro* images could be compared with our predicted infiltration measures to confirm heterogeneity or narrow the output space.

One way to quantify the infiltration of T cells is to determine the difference between the T cell mean and macrophage mean. The higher the value the more separation between the T cells and macrophages and, therefore, more structural heterogeneity within the spheroids. Spearman's rank correlation coefficients were calculated between this distance measure and outputs of interest at day 6 with $\alpha = 0.01$. Our model suggests this measure of separation between macrophages and T cells is not significantly correlated to total bacterial count ($\rho = 0.09$, $p = 0.07$). However, this model is only looking at runs that were calibrated to experimental data, which has a small range of bacterial count for the spheroid simulations. If increasing T cell separation was an isolated change it's possible that this relationship could be observed.

Although the bacterial counts in the spheroid simulations are all within a small range, the bacterial counts in the corresponding traditional simulations vary more. The Spearman's rank correlation coefficient between our separation measure and the traditional total bacterial count shows a significant positive correlation ($\rho = 0.56$, $p < 0.000001$). The parameter sets that show more separation in the spheroid have higher bacterial load in the traditional cultures compared to their paired spheroid simulations. This would suggest that those parameter sets rely on a lot of structure to be able to control bacteria, because when those parameters are used to simulate the traditional well-mixed conditions, the bacteria are not as well controlled. In contrast, those parameter sets that don't have a lot of separation, do equally well in controlling bacteria in both the spheroid and traditional. Parameter sets with less separation resemble the traditional organization, so similar results are expected. Thus, our simulations suggest that there are two different ways that the model can control the bacteria–one is structure dependent and the other is not. This aligns well with the experimental results as cells from the same patient can either 1) control bacteria in both the traditional and spheroid culture or 2) only control bacteria in the spheroid.

The spatial organization of cells within a spheroid also impacts aggregate macrophage and T cell activation. The separation measure positively correlates with activated macrophage count ($\rho = 0.31$, $p < 0.000001$), activated infected macrophage count ($\rho = 0.42$, $p < 0.000001$), total STAT1 macrophage count ($\rho = 0.31$, $p < 0.000001$), and total activated CD8+ T cells ($\rho = 0.17$, $p < 0.001$). Increasing separation is therefore associated with more macrophage, infected macrophage, and CD8+ T cell activation.

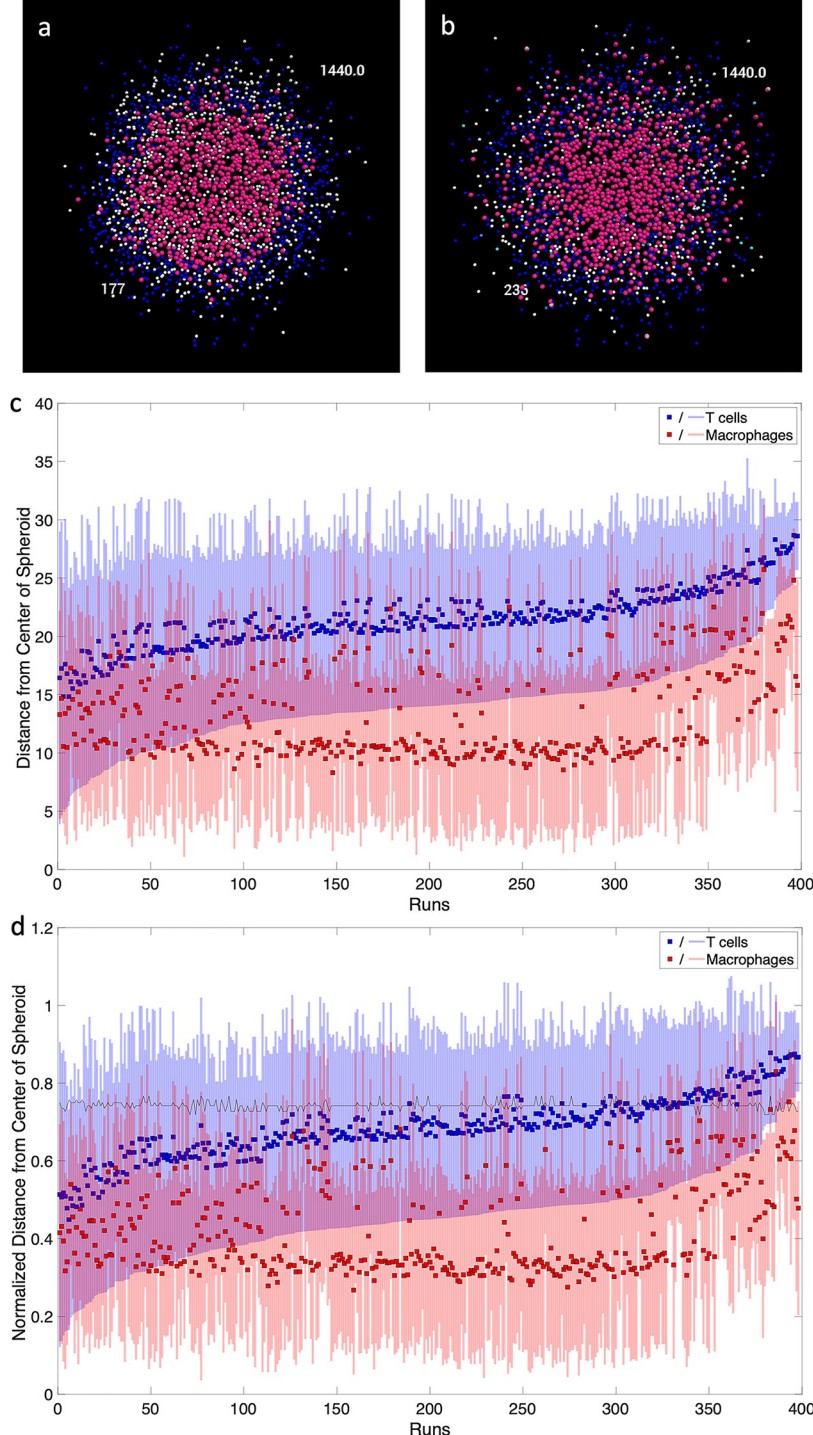

**Fig 6. T cell infiltration into macrophage core of spheroid.** Example simulations with a) extensive and b) limited T cell infiltration into the macrophage core. A slice through the center of the spheroid is shown at simulated day 6, or the 1440[th] six-minute time step. The number in the top right is the Repast time step, and the number in the bottom left is the simulation number. c) Mean and standard deviation radial density of macrophages and T cells (in numbers of grid squares). Runs are sorted by mean minus one standard deviation ($\mu-\sigma$) of the T cell distance from the center of the spheroid. d) Mean and standard deviation radial density of macrophages and T cells normalized to the initial size of the macrophage spheroid. The black line represents the initial size of the full spheroid, which varies due to differences in the *sphere efficiency*. Runs are sorted by mean minus one standard deviation ($\mu-\sigma$) of the normalized T cell distance from the center of the spheroid.

Taken together, these results illustrate that our simulations produce a wide range of outcomes that are consistent with the experimental data. Our findings suggest that differential immune mechanisms can control bacterial growth in a spheroid structure dependent or independent manner. Our results also support the hypothesis that spheroid structure enables important anti-bacterial immune mechanisms such as macrophage STAT1 activation and CD8+ T cell activation.

## Uncertainty analysis suggests that drug screening and macrophage activation mechanism investigations could yield different results between *in vitro* models

LHS-PRCC analysis is performed on the initial large LHS sweep to quantify how uncertainty in the parameters impacts uncertainty in the outputs of both the spheroid and traditional simulations. At day 2, before the T cells have been added to the simulation, the spheroid and traditional simulations show similar responses to changes in parameters (Fig 7). Total bacterial count is inversely correlated with the doubling time of the intracellular bacteria and the killing ability of the resting macrophages. This is expected since all the bacteria at the beginning of the simulation are intracellular. Lower doubling times of these bacteria, which is equivalent to faster growth rates, lead to more bacteria.

Before the addition of T cells, resting macrophages are responsible for all of the bacterial killing. Cytotoxic CD8+ T cells have yet to be added to the culture, and T cells are required to fully activate macrophages. *Mtb* killed by resting macrophages accounts for all of the killing and closely aligns with the total *Mtb* count. *Base killing probability* is therefore, as expected, negatively correlated with total *Mtb*.

Total NF-κB activated macrophages is the only other output, besides total bacterial count, showing significant correlations with parameters before day 2. NF-κB signaling comes from TNF-α or bacteria and is required for total activation. Total NF-κB activated macrophages are correlated to *NF-κB span*, *base killing probability*, *infected macrophage TNF secretion*, and *macrophage population max lifespan* in both spheroid and traditional simulations. *NF-κB span* is the length of time that NF-κB stays active after receiving the initial signal, so the longer this time period is, the more NF-κB activated macrophages there are. When *base killing probability* is lower, fewer bacteria are killed, and more bacteria and infected macrophages are available to activate NF-κB. Higher TNF-α secretion from infected macrophages also leads to more activation. Lastly, longer macrophage lifespans mean more macrophages are alive to be activated.

After the T cells are added, the total bacteria count is more dependent on CD4+ T cell parameters. The intracellular doubling time and base killing probability are both still negatively correlated with total bacteria count. The rest of the significantly correlated parameters are associated with CD4+ T cells and STAT1 activation. Fewer TB-specific CD4+ T cells, less CD4+ T cell activation, and more CD4+ deactivation all reduce the amount of activated CD4+ T cells indirectly leading to more bacteria. Higher threshold for STAT1 activation by IFN-γ, higher degradation rate for IFN-γ, and less CD4+ T cell IFN-γ secretion all lead to less macrophage STAT1 activation. Again, this will indirectly lead to more bacteria.

With the inclusion of adaptive immune cells, the responses of the spheroid and traditional simulations also diverge more. For bacterial counts, the only difference between spheroid and traditional simulations is due to external bacteria. Lower external doubling time leads to more bacteria only in the traditional simulation, as the population of external bacteria is so small in the spheroid simulation. Macrophage activation is mostly dependent on CD4+ T cell parameters in both models. However, increased macrophage activation in

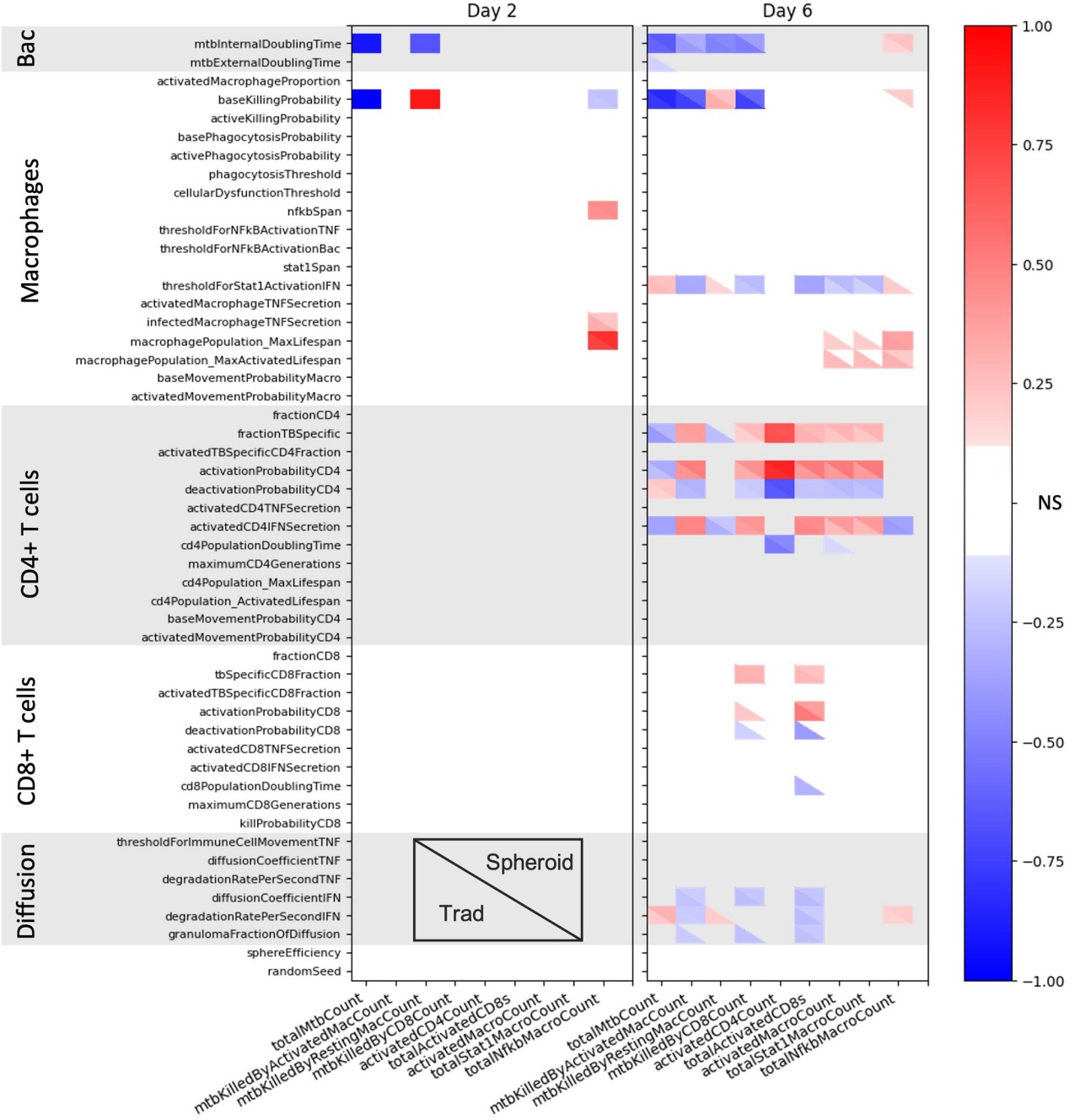

**Fig 7. Impact of input parameters on simulation outputs at day 2 before the T cells are added and day 6.** Correlation coefficients for spheroid and traditional simulations are shown in the same heatmap with the traditional in the lower left-hand corner and the spheroid in the upper right. Insignificant correlations are shown in white, while positive and negative correlations are shown with red and blue, respectively. Significance was determined with α = 0.01 and a Bonferroni correction.

the traditional simulation is also correlated with increased macrophage lifespans and decreased CD4+ T cell doubling time. This suggests that macrophage death might be limiting the population size in traditional runs. Also, a larger CD4+ T cell population will lead to more macrophage activation.

CD8+ T cell activation is correlated with parameters related to CD4+ T cells, CD8+ T cells, and IFN-γ. Differences between the two models include negative correlations between the probability of deactivation/population doubling time and total CD8+ T cell activation in the traditional simulations only. Less deactivation or more proliferation should therefore lead to higher activated CD8+ T cell populations in the traditional simulations.

The only parameter-output relationship seen just in the spheroid simulation is a positive correlation between the *base killing probability* and total NF-kB activated macrophages. This is the opposite of the relationship seen at day 2. One hypothesis for this relationship is that more killing initially leads to less macrophage activation and subsequent death later in the simulation. This is supported by positive correlations between macrophage lifespans and total NF-kB activated macrophages.

Despite these differences, the two models have many similar responses to changing parameters. For example, the total activated CD4+ T cells show the same relationships for both the spheroid and traditional simulations with regard to all parameters. The significantly correlated parameters are all related to CD4+ T cells: fraction of TB-specific cells, activation probability, deactivation probability, and doubling time.

Altogether, these results show that similar parameters are driving dynamics in the spheroid and traditional models before day 2, but the influential parameters diverge after the addition of T cells. We hypothesize that this diversion in influential parameters is due to the differing relative location of the T cells to macrophages in traditional versus spheroid simulations. These correlations can inform which *in vitro* model is most appropriate when designing experiments. For example, the traditional simulation has a correlation between macrophage lifespans and macrophage activation that is not seen in the spheroid (i.e., *macrophage population max lifespan* and *macrophage population max activated lifespan* in Fig 7). This relationship suggests that macrophage lifespans influence macrophage activation in the traditional culture, so a spheroid might be more appropriate if the biological question under investigation relates to the drivers of macrophage activation. The traditional simulation also has an inverse correlation between external bacterial doubling time and total bacterial count, which suggests more of the bacterial population is extracellular. This may lead to overestimation of drug efficacy in screenings as intracellular bacteria may receive sub-therapeutic levels of antibiotics. Finally, there is a correlation between IFN-γ parameters and resting macrophage killing in the traditional simulation (i.e., *threshold for STAT1 activation IFN and degradation rate per second IFN)*, which suggests less macrophage activation and more resting macrophage killing. IFN-γ treatment or treatment with other immunotherapies could therefore have different impacts in spheroid vs. traditional cultures. Our results therefore indicate that spheroid and traditional cultures could have differential responses in relation to both mechanism investigations and drug screenings.

## Limitations in representing both spheroid and traditional cultures guide future model iterations

The analysis done thus far is based on paired calibration. Pairing the simulations assumes that everything except for the initialization and movement is the same between the spheroid and traditional simulations. While this assumption allows us to reproduce most of the experimental range, the highest traditional culture CFU counts are unable to be recreated with paired simulations. Traditional simulations with high levels of bacteria falling in this range are observed, but the corresponding spheroid simulation did not meet calibration criteria. This suggests that to reproduce these high traditional CFU results, some parameters (i.e., biological mechanisms) may need to be different between the spheroid and traditional simulations. To

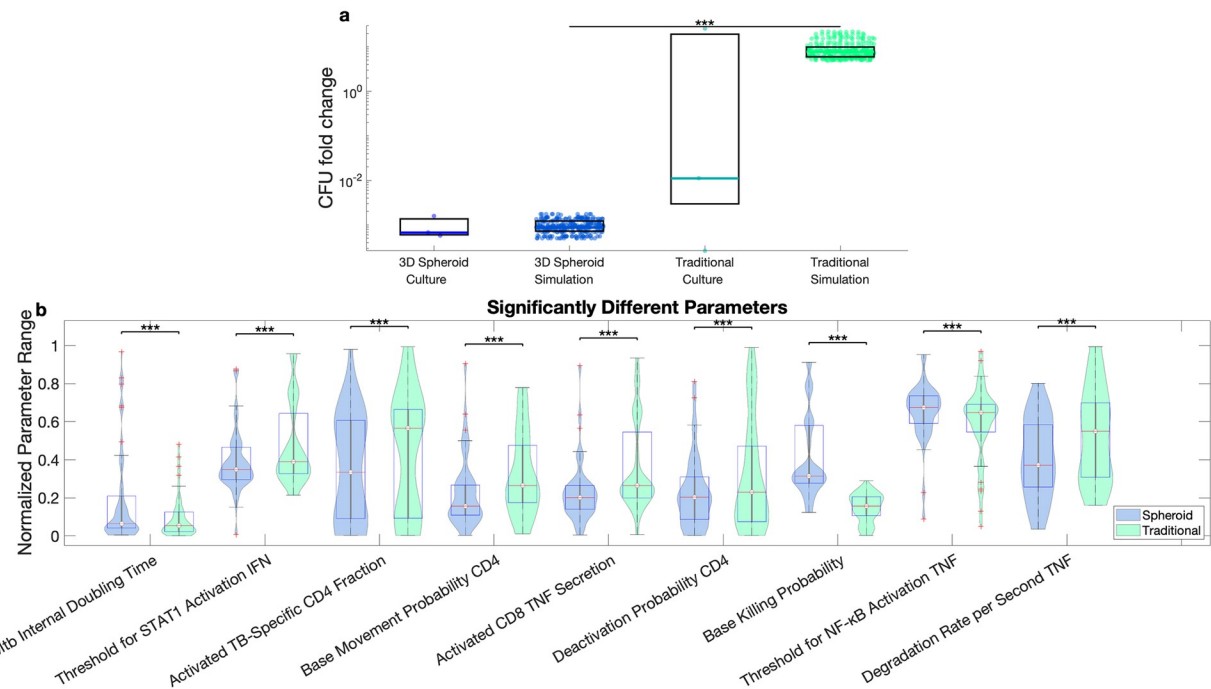

**Fig 8. Matched spheroid and traditional simulations.** a) Traditional simulations that fell above the paired range and matched spheroid simulations. b) Significantly different parameters between the set of high traditional simulations and matched spheroid simulations.

investigate this possibility, we evaluate unpaired simulations that are allowed to have different parameter values between traditional and spheroid simulations, but that are matched as closely as possible for initial conditions.

Comparing the parameters between the spheroid and traditional simulations, nine parameters are found to be significantly different between the matched traditional and spheroid simulations (Fig 8). Some of the observed parameter differences can lead to fewer bacteria in the spheroid directly or indirectly by increasing activation. The matched spheroid runs had higher intracellular doubling time of the bacteria meaning the bacteria grow more slowly and a higher resting macrophage killing rate leading to more bacterial killing as compared to the traditional simulations. Therefore, the matched spheroid runs have fewer bacteria than the traditional runs due to less intracellular bacterial growth and more killing, suggesting that macrophages might be more efficient and inhibiting bacterial growth and killing bacteria in the spheroid conditions.

The matched spheroid runs also have parameter differences that lead to more macrophage activation. Lower *threshold for STAT1 activation IFN* in spheroid runs would give more STAT1 activation of macrophages causing more overall macrophage activation. Lower *deactivation probability CD4* in spheroid runs would prolong CD4+ T cell activation giving these cells more opportunities to activate macrophages. Lastly, lower *degradation rate per second TNF* in spheroid runs maintains higher concentrations of TNF-α, leading to more NF-κB activation of macrophages.

The roles of the other parameters are less clear. Lower *activated TB-specific CD4 fraction*, lower *activated CD8 TNF secretion*, and higher *threshold for NF-κB activation TNF* would all suggest lower macrophage activation in the spheroid. Lower *base movement probability CD4*

could delay activation of TB-specific T cells or could lead to less spatial interference by non-TB-specific T cells in spheroids. As these simulations are matched after the fact, some of these differences are potentially spurious.

These predicted differences could be tested *in vitro*. For example, bacterial metabolism, markers of macrophage activation, and markers of CD4+ T cell activation could be measured and compared between the spheroid and traditional cultures. Bulk RNAseq already suggests augmented T cell activation in spheroid cultures [15]. These model-predicted differences also guide future computational and experimental studies by highlighting hypothesized cellular functional differences between traditional and spheroid cultures.

## Discussion

*In silico* models have been used previously to represent multiple *in vitro* models for other diseases. In 2006, Grant et al. used cellular automata to represent the growth of epithelial cells in 4 conditions: 3D embedded, suspension, surface, and collagen overlay cultures [68]. They were able to recreate the complex structure associated with each condition with a set of axioms governing the interactions of cells, matrix, and cell-free space. The difference between a 2D and 3D culture system has also been modeled to explore viral dynamics and drug toxicity. A network model of tumor cell infection by oncolytic viruses was simulated in a 2D monolayer and 3D environment, which suggested that traditional mean field models overestimate how effective therapy would be [69]. Beyond this, infection in a 3D environment was shown to have a smaller chance of tumor eradication, emphasizing the need for ideal virus characteristics: fast replication and slow tumor cell killing. A virtual cell-based assay was extended from 2D cultures to 3D spheroids to predict drug toxicity [70]. This model was found to represent 3D *in vitro* models well, which show higher drug toxicity than 2D monolayers. In silico models have further been used to compare *in vitro* models with *in vivo* models. Hamis et al. showed that *in vivo* responses of LoVo cells to an anti-cancer drug (AZD6738) could be qualitatively predicted using an *in silico* spheroid model with parameters determined using an *in vitro* monolayer [71].

In all instances mentioned above, space is explicitly modeled to gain insight into the system behavior in different configurations. The spatial configurations alter the dynamics of the system and change important predictive outcomes, such as drug response. Similar to these prior works, we explicitly include space to model two different *in vitro* culture conditions. We show that spatial organization alone can change the dynamics of *Mtb* infection *in vitro*. We use our simulations to predict which model outcomes are likely to be affected by spatial organization (e.g., macrophage activation and bacterial killing) and thereby guide experimental decisions.

Here we are using a single *in silico* framework to represent separate traditional and spheroid cell cultures. A separate problem, representing one *in vivo* or *in vitro* system with both a 2D and 3D computational model, has also been addressed [72,73]. Models of *in vivo* granulomas and *in vitro* spheroids suggest that 2D representations of 3D systems (i.e. slice through center of structure) have similar results and save computational time [72,73].

Granulomas are spatially organized structures, with a core of macrophages and a cuff including CD4+ and CD8+ T cells [74]. The center of the granuloma is a more pro-inflammatory environment, while the cuff has more anti-inflammatory cytokines [75,76]. Higher frequencies of pro-inflammatory cytokines or lymphocytes are correlated with lower bacterial burden in granulomas, but it is suggested that a balance of pro- and anti-inflammation is necessary to limit both bacterial growth and pathology [76–78]. While we don't explicitly include anti-inflammatory pathways in this preliminary model, our model predictions are well-aligned with observed patterns of pro-inflammatory signaling within granulomas. Specifically,

macrophage activation in our simulations is limited by the interactions between IFN-γ and macrophages which begins at the periphery of the macrophage core and moves inwards. This result aligns with the observed pattern of p-STAT1 seen in peripheral regions of granulomas in immunohistochemistry of NHP granulomas [79]. Previous computational modeling also suggests the importance of IFN-γ producing T cells and interactions between macrophages and T cells for bacterial control [80]. It emphasizes the importance of spatial organization as interactions between CD11c+ macrophages and T cells are limited due to the cellular distributions within granulomas and the recruitment of non-specific T cells. While the structure of our spheroid simulations is artificially constructed rather than emerging from immune interactions, the similar spatial patterns for cell organization and activation suggest that the simulation reproduce this key aspect of granuloma organization.

The *in vitro* cultures we are simulating primarily represent cellular granulomas, rather than caseous or necrotic granulomas. Necrotic granulomas are important drivers of active disease and make treatment challenging as many drugs struggle to penetrate into the necrotic cores of these granulomas [81]. While our *in vitro* spheroids don't have a fully necrotic core, they do show early markers of necrosis [15]. Nonnecrotic and necrotic alveolar macrophages were seen in the center of the spheroid, as shown with the cytoplasmic localization of HMGB1. This necrosis was seen without experimentally induced hypoxia, which was necessary for necrosis in traditional culture. This spheroid culture and our *in silico* model thus would have utility for drug screenings mainly for cellular granulomas, but also as an early-stage necrotic granuloma. Screening for drug candidates in *ex vivo* caseum or caseum surrogates could be paired with spheroid cultures to ensure efficacy in the necrotic core in addition to the spheroid structure [81,82].

The evolution of a single granuloma can be followed over time in other experimental systems. Sequential imaging with [18F] fluorodeoxyglucose positron emission tomography and computed tomography has been used to follow disease progression in NHP and track response to TB treatment in humans [83–85]. This imaging gives information at the lesion-to-tissue scale. Florescent *in vivo* microscopy of zebrafish embryos has given insight into the cellular level dynamics [86]. Imaging after infection of zebrafish embryos with *Mycobacterium marinum* allows tracking of infected macrophages providing information about early granuloma formation and dissemination. Recently, a method to study zebrafish granulomas *ex vivo* called Myco-GEM was created that allows continuous light sheet imaging for upwards of 8 hours [87]. With the tagging of cytokines, specific cells, or bacteria the inflammatory state of the granuloma, granuloma dynamics, cell movement, and bacterial load can be longitudinally examined. Our computational model similarly provides dynamic information at the cellular scale. Beyond this, we gather information about aggregate cell counts, cell activation status, and cytokine concentrations without perturbing the observed system. Thus, our computational model complements *in vitro* experimental models, by providing both high-resolution spatiotemporal information and aggregate information about host-pathogen interactions within individual granuloma structures. Simulations with virtual perturbations and knockouts can then quickly be run to examine how these interactions could contribute to bacterial survival or elimination.

TB is a very heterogeneous disease. There are many different clinical outcomes: bacterial clearance, asymptomatic latent infection, and active infection [78,88]. These host-level outcomes are dependent on a population of granulomas, which can be very heterogeneous even within the same lung leading to bacterial dissemination, control, or clearance [78,85,88,89]. Our *in vitro* models are more controlled with an established structure and proportion of cell types. Nonetheless, the *in vitro* model still showed a large range of bacterial control, which we recreated *in silico*. We also see heterogeneity in T cell localization *in silico*. While this is not

seen as much *in vitro*, there is some variability *in vivo*. Early granulomas have T cells dispersed throughout, while well-developed ones are more structured with a ring of T cells on the periphery [90]. Being able to reproduce a diversity of granuloma organizations will allow us to explore how different microenvironments contribute to granuloma trajectory and treatment response.

LHS-PRCC has been used to look at correlations between inputs and outputs in simulations of *in vivo* NHP granulomas. While our time points don't line up with the longer *in vivo* simulations, we can compare parameter influences before and after adaptive immunity has been added. In the first iteration of the NHP granuloma simulation, there are similarities to our model. This model from literature shows a strong positive correlation between intracellular growth rate and total extracellular bacteria during early infection [18]. As infection progresses extracellular bacteria in the simulated NHP granulomas become negatively correlated with T cell parameters, namely recruitment, movement, and activation of macrophages.

Our primary output of interest, total bacteria, shows a similar relationship with the intracellular growth rate before the addition of the adaptive immune system. Some comparisons between this *in vivo* simulation and our *in vitro* simulation are limited because *in vivo* mechanisms are missing *in vitro*, like cellular recruitment. However, we see an increased importance of T cell parameters on our output of interest after adaptive immunity is initiated as in this literature model. In our short-term assembled spheroids, we see similar parameter influences to *in vivo* granulomas. Therefore, some comparisons can be made not only between our two *in vitro* models, but also *in vivo* simulations, to be able to rationally identify good use cases for various *in vitro* models.

Our model is not without limitations. The only PBMC-derived CD3+ T cells simulated are CD4+ and CD8+ T cells. Some subsets of T cells (e.g., regulatory and γδ T cells) are excluded from the model for the purpose of simplification. Simplifications are also made to the macrophage activation pathway. The model only incorporates M1 macrophage polarization/activation represented as a 2-step pathway, and M2 macrophage polarization is not included. Additionally, phenotypic variance of the bacteria was left out, even though there is a known persistence of slow-growing *Mtb* in response to environmental stresses [22]. While many of the parameters are varied between all cells within a population, some are held constant across the entire population of cells. This model construction choice was made for simplicity but could be changed in the next model iteration. Cellular heterogeneity could be represented with more detail by varying secretion rates, maximum generations for T cells, and thresholds for activation, movement, and cellular dysfunction. Varying secretion rates would better represent the distribution of secretion by immune cells [39]. Xue et al. 2015 suggest that only 5% of cells are responsible for 60% of all TNF-α production [39]. We expect that the inclusion of this heterogeneity would further highlight the importance of spatial layout as there would be a smaller population of cells secreting cytokines. We expect that changes in generation size or thresholds would increase variance in outputs without changing the average behavior. Additionally, our model has been calibrated to be used with cells from patients with presumed active TB. The same cells derived from an uninfected patient or a patient with latent TB might behave differently, and the model would need to be recalibrated to different data. These assumptions can be reassessed as we iterate this model to use it in answering new biological questions.

While our *in vitro* model can represent a majority of the characteristics that could be incorporated into a complex *in vitro Mtb* model, it still diverges from the idealized. No explicit environmental impact on the cells in the simulation is included. It's known that plastic and glass plates differ from *in vivo* environments, and as such extracellular matrix (ECM) components like collagen have been incorporated into *in vitro* models. ECM can also change the lifespan and movement of the cells and sequester chemokines. We plan to incorporate ECM in future

iterations. While primary human cells were represented, the bacteria represented within this model is BCG, a model organism for *Mtb*, rather than *Mtb* itself. BCG was used for preliminary analysis as it can be used outside of a BSL3 laboratory. Switching between BCG and *Mtb* could be done by adjusting parameter values, but more detailed pathways would need to be added if specific virulent strains were of interest.

## Conclusion

In summary, we show a novel application of ABMs to *in vitro* TB infection cultures. In doing so, we introduce a framework to potentially integrate results from and compare multiple *in vitro* models. Our findings identified the characteristics in which traditional and spheroid cultures differ, and how these *in vitro* models could be further used to narrow down biologically important mechanisms. We find that the timing of macrophage activation is limited by T cell interactions and macrophage location, that bacteria can be controlled in a structure dependent or independent manner, and that certain investigations (e.g., drug screening and macrophage activation mechanism investigations) could produce different results in the traditional and spheroid models.

## Supporting information

**S1 Fig. Distribution of parameters in calibrated runs.** The ranges of the parameters have been normalized from 0 to 1 with the bounds representing the minimum and maximum of the ranges listed in *Table 1*.
(TIF)

**S2 Fig. Scaled-up simulations.** Spheroids that have been scaled to 1/2, 1/5, 1/10 (original), 1/20, and 1/50 size of the experimental culture were simulated using the calibrated parameters. Outputs for the scaled spheroid and traditional simulations were compared with the 6 outputs used for calibration: spheroid CFU change with zoomed y-axis (a), traditional CFU fold change (b), spheroid cell viability (c), traditional cell viability (d), spheroid cell count multiplied by one over scaling factor (e), and traditional cell count multiplied by one over scaling factor (f). Red regions represent the experimental ranges. As expected, the outcomes of the traditional simulation are similar regardless of how much it was downscaled. The spheroid simulation showed similar outcomes for normalized cell count and percent viability, but CFU fold change varied. Smaller simulated spheroid had lower CFU fold changes suggesting they are better able to control bacteria. These dots represent 398 runs, but one run is missing from the 1/2 spheroid simulation population and three runs are missing from the 1/2 traditional simulation population due to these runs exceeding wall time limits with maximum memory and time allocated.
(TIF)

**S3 Fig. Scaled-up simulations that maintain CFU fold changes.** Subset of simulations that fall within spheroid and traditional CFU fold changes for both 1/2 and 1/10 scaled runs. CFU fold changes for spheroid simulations (a) and traditional simulations (b). Red regions represent experimental ranges. Time courses of bacteria count for traditional (lighter green) and spheroid (darker blue) show similar dynamics across scales (c-g).
(TIF)

**S4 Fig. Average radial distribution.** The radial distribution of a) macrophages, CD4+ T cells, and CD8+ T cells; b) base and activated T cells; c) NF-κB and STAT1 activated macrophages; d) base and activated macrophages. The y-axes represent the radial density of cells, which is calculated by number of cells at a given distance from the center of a spheroid divided by the

volume of the spherical shell. All runs have been averaged with error bars representing standard deviation.
(TIF)

**S5 Fig. Flowchart of the simulation.** After initialization, the simulation consists of a loop of agent secretion, diffusion, agent behaviors, and an agent watcher. Overview of actions is shown, and more detail can be found in the Repast model code at https://github.itap.purdue.edu/ElsjePienaarGroup/TB-in-vitro release v1.0.1.
(PDF)

**S1 File.**
(DOCX)

## Acknowledgments

We thank Lev Gorenstein and the rest of the Research Computing Staff for their assistance with batch computing at the Rosen Center for Advanced Computing. We would also like to acknowledge Catherine Weathered for her mentorship and her work setting up the simulation foundations in Repast and Slurm.

## Author Contributions

**Conceptualization:** Elsje Pienaar.

**Data curation:** Leigh Kotze.

**Software:** Alexa Petrucciani, Alexis Hoerter.

**Supervision:** Nelita Du Plessis.

**Visualization:** Alexa Petrucciani.

**Writing – original draft:** Alexa Petrucciani.

**Writing – review & editing:** Alexis Hoerter, Leigh Kotze, Nelita Du Plessis, Elsje Pienaar.

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
