## [Decision Letter · Decision Letter 0]

22 Nov 2023

PONE-D-23-27966In silico agent-based modeling approach to characterize multiple in vitro tuberculosis infection modelsPLOS ONE

Dear Dr. Pienaar,

Thank you for submitting your manuscript to PLOS ONE. After careful consideration, we feel that it has merit but does not fully meet PLOS ONE’s publication criteria as it currently stands. Therefore, we invite you to submit a revised version of the manuscript that addresses the points raised during the review process.

We look forward to receiving your revised manuscript.

Kind regards,

Bhanwar Lal Puniya, Ph.D.

Academic Editor

PLOS ONE

Journal Requirements:

This work used the Extreme Science and Engineering Discovery Environment (XSEDE), which is supported by National Science Foundation grant number ACI-1548562. Anvil at Purdue and Expanse at UCSD were used through allocation TG-MDE220002. This research was done using services provided by the OSG Consortium (87–89), which is supported by the National Science Foundation awards #2030508 and #1836650. We also thank Lev Gorenstein and the rest of the Research Computing Staff for their assistance with batch computing at the Rosen Center for Advanced Computing. We would also like to acknowledge Catherine Weathered for her mentorship and her work setting up the simulation foundations in Repast and Slurm.

4. Please remove your figures from within your manuscript file, leaving only the individual TIFF/EPS image files, uploaded separately.

Reviewers' comments:

Reviewer's Responses to Questions

**Comments to the Author**

1. Is the manuscript technically sound, and do the data support the conclusions?

Reviewer #1: Yes

Reviewer #2: Yes

2. Has the statistical analysis been performed appropriately and rigorously? 

Reviewer #1: Yes

Reviewer #2: Yes

3. Have the authors made all data underlying the findings in their manuscript fully available?

Reviewer #1: Yes

Reviewer #2: Yes

4. Is the manuscript presented in an intelligible fashion and written in standard English?

Reviewer #1: Yes

Reviewer #2: Yes

5. Review Comments to the Author

Reviewer #1: The paper describes an agent-based approach for modeling and simulating the Mycobacterium tuberculosis (Mtb) infection. It compares interactions between immune cells and bacteria in two types of in vitro models: the traditional monolayer culture and a biomimetic 3D spheroid granuloma developed by the authors in previous a previous study. The authors leveraged the capability of ABMs to tackle 3D interactions to quantify the heterogeneity of the host response within and between different spatial organizations.

Even if the simulations were performed at a reduced spatial scale and for a limited time, they clearly adhere to the experimental results and are interestingly predictive of some qualitative and quantitative features of the corresponding in vitro models. The provided results are presented appropriately and supported by data.

However, the article has its flaws, mainly in the model presentation.

My biggest concern regards a choice in the model description that affects mainly the Methods section but also part of the Results. In the article’s main text, a simple name or an acronym would be preferred to identify a variable and discuss its role rather than using its code “lowerCamelCase”; this is a programming good practice that, in my opinion, cannot be translated directly into scientific texts. The technical details of the model (especially those of Sections 2.2.1 and 2.2.2) can be provided in the Supplementary Information, accompanied by a proper model specification (algebraic or through UML diagrams) currently missing in the article. At least, the model's variables and parameters should have been introduced and contextualized before, as SI Table 1, which provides values and references for them, is mentioned only in Section 2.2.4. To improve the Methods section, only the key aspects of the model needed to support the scientific results should be maintained. The general modeling approach is well-described and synthesized through Figures 1 and 2, a good starting point later abandoned to dive too much into technical details.

I also question the choice of using Repast Simphony, which could not handle the simulation’s computational burden and forced the authors to approximations (e.g., simulating 1/10th of the actual spheroid sizes and limiting the number of cytokines). Shouldn’t using Repast HPC or a completely different ABM framework (e.g., DMASON or FLAME GPU) be better if the computational cost of the simulations was excessive for Repast Simphony?

I couldn’t find the Repast model through the provided Zenodo links. Is there a dedicated Git repository?

Finally, grammar (in particular, punctuation, articles, and verbs) should be double-checked across the entire document, perhaps with the help of an automated tool; I found several errors and typos. To report a few:

- L68-69: a comma before “which” and one after “e.g.” are required.

- L548: “aligns” should be “align.”

- L550 “there are” should be “there is”; “at day 6” should be “on day 6.”

- L558: Missing comma to enclose the adverbial phrase: […] that, in most situations, the [...]

- L563: “that that”.

- L581: “indicate” should be “indicates.”

- L605: “follow” should be “follows.”

- L867: “represents” should be “represent.”

- Check the use of heterogenous vs. heterogeneous across the article (https://grammarist.com/usage/heterogeneous-heterogenous/)

In general, a little more conciseness would benefit the article’s readability.

MINOR QUESTIONS AND REMARKS:

L57: “This computational tool is used to help integrate data from different laboratory models and generate hypotheses about TB disease to be tested in laboratory models.” This sentence could be rewritten for more clarity.

L194: CFU was never defined before (I think no acronym should be taken for granted, even common ones).

L250: “The PDE is run with a smaller time step than the ABM, ranging from 2 to 14 diffusion iterations per agent time step, depending on the diffusion parameters.”

What modeling solution has been adopted so that this choice does not affect the consistency of the model?

L615: “These distributions can be tested by staining and imaging slices of spheroid at dynamic time points. We recommend that activation markers for macrophages be spatially analyzed on days 3 and 4.”

Taking this sentence as it is, it looks like the authors entrust the reader with model validation.

L911-915: (18) is cited three times consecutively; one is sufficient.

Reviewer #2: In this paper the authors simulate in vitro Tb infection models, spheroids and traditional cell culture. They developed an ABM to compare the simulations from these models and understand the mechanistic differences between the framework using: uncertainty analysis. They demonstrate and present a qualitative comparison of their simulation results from the spheroid and traditional cultures to that of the experimental results. The paper is well written, however there are some points and sub figures that need clarity. Attached are suggestions that can help enhance clarity.

In addition, authors are encouraged to upload the model files.

6. PLOS authors have the option to publish the peer review history of their article (what does this mean?). If published, this will include your full peer review and any attached files.

Reviewer #1: No

Reviewer #2: **Yes: **Meghna Verma

---

## [Author Response · Author response to Decision Letter 0]

18 Dec 2023

See attached document, copied below for completeness.

Comments from reviewers:

1. Reviewer #1: The paper describes an agent-based approach for modeling and simulating the Mycobacterium tuberculosis (Mtb) infection. It compares interactions between immune cells and bacteria in two types of in vitro models: the traditional monolayer culture and a biomimetic 3D spheroid granuloma developed by the authors in previous a previous study. The authors leveraged the capability of ABMs to tackle 3D interactions to quantify the heterogeneity of the host response within and between different spatial organizations.

Even if the simulations were performed at a reduced spatial scale and for a limited time, they clearly adhere to the experimental results and are interestingly predictive of some qualitative and quantitative features of the corresponding in vitro models. The provided results are presented appropriately and supported by data.

However, the article has its flaws, mainly in the model presentation.

1.1. My biggest concern regards a choice in the model description that affects mainly the Methods section but also part of the Results. In the article’s main text, a simple name or an acronym would be preferred to identify a variable and discuss its role rather than using its code “lowerCamelCase”; this is a programming good practice that, in my opinion, cannot be translated directly into scientific texts. 

We agree that the camel case is more of a coding convention than manuscript convention. We have changed how the parameters are referred to in the text throughout, so they no longer follow camel case formatting. They are now simple names that clearly identify their meaning and are italicized to show they are model parameters. We have updated Fig 8 to reflect this change. Fig 7 was left as is due to the density of text, and to support readability, especially in the diagonal labels.

1.2. The technical details of the model (especially those of Sections 2.2.1 and 2.2.2) can be provided in the Supplementary Information, accompanied by a proper model specification (algebraic or through UML diagrams) currently missing in the article. 

We have edited the methods sections to reduce the number of unnecessary details. However, given that this is the first publication of this model, we believe that some technical detail of the model belongs in the main text. We believe that this is necessary to support a) clarity of the results sections when the impact of specific mechanisms is discussed and b) reproducibility for computational readers who might want to reimplement parts of the model. By keeping these model details within the main text, it also better allows us to respond to other reviewer comments that ask for more clarifications (e.g. comments 1.3, 1.12, 2.2, 2.3, 2.5)

We appreciate the reviewer’s suggestion of including a UML diagram as this will further support reproducibility. We have added a flowchart of the model structure to the supplement as S5 Fig, a simplified UML activity diagram.

1.3. At least, the model’s variables and parameters should have been introduced and contextualized before, as SI Table 1, which provides values and references for them, is mentioned only in Section 2.2.4. 

This is an excellent point that variables and parameters were not clearly introduced early in the methods. The calibrated variables are given in the main text Table 1. We moved this table up to the beginning of the Model structure section to help contextualize the parameters. S1 Table only provides values and references for the variables kept constant. 

1.4. To improve the Methods section, only the key aspects of the model needed to support the scientific results should be maintained. The general modeling approach is well-described and synthesized through Figures 1 and 2, a good starting point later abandoned to dive too much into technical details.

As mentioned in our response to comment 1.2 above, we have shortened the methods, focusing our efforts on the sections for Diffusing molecules and Agents.

1.5. I also question the choice of using Repast Simphony, which could not handle the simulation’s computational burden and forced the authors to approximations (e.g., simulating 1/10th of the actual spheroid sizes and limiting the number of cytokines). Shouldn’t using Repast HPC or a completely different ABM framework (e.g., DMASON or FLAME GPU) be better if the computational cost of the simulations was excessive for Repast Simphony?

This is a valid concern, and we thank the reviewer for their suggestions. These are good tools that we will continue to investigate as we grow our model. Currently, in our efforts at reducing computational cost, a code profiler identified that our PDE model of 3D diffusion is the most computationally expensive and limiting part of the simulation. For this reason, we are not sure switching to DMASON or FLAME GPU will be beneficial to reduce the computational requirements for 3D diffusion, as neither of them appear to have built in diffusion, so the same custom diffusion solvers would have to be coded in those systems. 

1.6. I couldn’t find the Repast model through the provided Zenodo links. Is there a dedicated Git repository?

Thanks for noting this omission. The Repast model code is located at https://github.itap.purdue.edu/ElsjePienaarGroup/TB-in-vitro release v1.0. This has been added to the Simulation and parameter sampling section. 

1.7. Finally, grammar (in particular, punctuation, articles, and verbs) should be double-checked across the entire document, perhaps with the help of an automated tool; I found several errors and typos. To report a few:

- L68-69: a comma before “which” and one after “e.g.” are required.

- L548: “aligns” should be “align.”

- L550 “there are” should be “there is”; “at day 6” should be “on day 6.”

- L558: Missing comma to enclose the adverbial phrase: […] that, in most situations, the […]

- L563: “that that”.

- L581: “indicate” should be “indicates.”

- L605: “follow” should be “follows.”

- L867: “represents” should be “represent.”

Thank you for your detailed review, we have made the adjustments as suggested.

Also, we used an automated grammar checker to check the entire document. 

1.8. Check the use of heterogenous vs. heterogeneous across the article (https://grammarist.com/usage/heterogeneous-heterogenous/)

Thank you for pointing this out, we have corrected all instances to heterogeneous.

1.9. In general, a little more conciseness would benefit the article’s readability.

We have edited for conciseness throughout the manuscript.

MINOR QUESTIONS AND REMARKS:

1.10. L57: “This computational tool is used to help integrate data from different laboratory models and generate hypotheses about TB disease to be tested in laboratory models.” This sentence could be rewritten for more clarity.

We have split this sentence into two to help with clarity.

1.11. L194: CFU was never defined before (I think no acronym should be taken for granted, even common ones).

We defined colony-forming unit as CFU at its first mention, thanks for this catch.

1.12. L250: “The PDE is run with a smaller time step than the ABM, ranging from 2 to 14 diffusion iterations per agent time step, depending on the diffusion parameters.”

What modeling solution has been adopted so that this choice does not affect the consistency of the model?

The modeling solution we use is the stability criterion, which allows the PDE time step and number of iterations to be calculated in a way that does not affect the consistency of the model. This process has been clarified in the text to indicate PDE time step and number of diffusion iterations are determined using the stability criterion and diffusion coefficients as previously described for agent-based models. (Cilfone, Kirschner, and Linderman 2015)

1.13. L615: “These distributions can be tested by staining and imaging slices of spheroid at dynamic time points. We recommend that activation markers for macrophages be spatially analyzed on days 3 and 4.” Taking this sentence as it is, it looks like the authors entrust the reader with model validation.

We have made this recommendation into an example to remove the sense that validation is dependent on reader. We are planning to do more experimental work with the spheroids.

1.14. L911-915: (18) is cited three times consecutively; one is sufficient.

Removed two citations.

2. Reviewer #2: 

In this paper the authors simulate in vitro Tb infection models, spheroids and traditional cell culture. They developed an ABM to compare the simulations from these models and understand the mechanistic differences between the framework using: uncertainty analysis. They demonstrate and present a qualitative comparison of their simulation results from the spheroid and traditional cultures to that of the experimental results. The paper is well written, however there are some points and sub figures that need clarity. Attached are suggestions that can help enhance clarity.

2.1. Authors are recommended to add what a monolayer/traditional culture in vitro model is comprised of. How are these different than spheroids both in terms of structure, constituents, and complexity. 

We have added a sentence to the introduction clarifying that the traditional monolayer culture is comprised of the same cells and cell numbers, from the same donor, but with no structure or magnetic nanospheres.

2.2. What is the maximum age of the cell and how it differs from death rate used otherwise. Per 2.2.4, maximum generations are held constant between all cells, does that relate to the age of the cells and the rates used? 

We believe that this uncertainty stems from our insufficient distinction between maximum age of a single cell vs. the maximum cell generations that single cells can produce through proliferation.

Maximum age of the cell is used to determine when cells die. Cells age either at the base aging rate or activated aging rate, which are calculated by individual resting and activated lifespans. When the cumulative level of aging reaches the maximum age of a cell, then the cell dies. Therefore, we don’t have cell death implemented as a death rate for individual cells, but rather cell death is determined by an aging parameter.

In contrast, the parameter for maximum generations determines how many generations of daughter cells can be produced from a single activated proliferating cell. The cell age will reset to zero for new daughter cells.

This separation between maximum age and maximum generation is now clarified in our new SI Figure 5, and we have clarified in the Simulation and parameter sampling section that the maximum generations parameter relates to the number of times a cell can proliferate.

2.3. L293: What are the set durations for the NF-kB and STAT1 activation and how are these determined? 

The set durations for the NF-κB and STAT1 were sampled between 0.16 and 166 as given in Table 1. These values were derived from parameters determined in previous computational models as cited in Table 1. 

2.4. L: 306: Explain what within an infected macrophage mean? Do the authors mean from the infected macrophage’s Moore neighborhood? 

We clarified that each infected macrophage corresponds to a list of one or more intracellular bacteria that exist in the same location on the corresponding bacterial grid.

2.5. What is the threshold of intracellular bacteria in monolayer versus spheroid structures, are these assumed to be the same i.e. 20-40 from a uniform distribution? 

Yes, this is kept the same. We have clarified in Initial conditions section that all parameters are kept constant between monolayer and spheroid simulations except those dictating structure and gravity-limited movement (case number and is gravity in S1 Table).

2.6. For Fig 4, to perform a head on comparison to ‘4f’, does 4g time step of 6min correspond to day 6? Please explain. 

We have clarified that this is at simulated day 6, which is at the 1440th 6-minute time step. So, this figure is a direct comparison of the structures at day 6. 

2.7. Define what sphere efficiency is and the formula that defines it. 

Sphere efficiency is the fraction of grid cubes in the spheroid that are occupied by cells, this can be thought of as the sum of the volume of all cells divided by the volume of the spheroid at initialization. The definition and formula can be found on lines 380-384 (421-425 in the tracked version).

2.8. What do 173 and 233 in Fig 6a represent? Are these the number of cells at Day 6? 

The number in the top right is the Repast time step, and the number in the bottom left is the simulation number. This has been clarified in the figure caption.

2.9. L 755-756: Why do T cell addition lead to diversion in influential parameters in traditional versus spheroidal? Can the authors suggest a hypothesis if possible? 

We hypothesize that this diversion in influential parameters is due to the differing relative location of the T cells to macrophages in traditional versus spheroidal. This hypothesis has been added to the manuscript and is explored in more detail in further work currently in preparation. 

2.10. The legend for Fig 8b looks similar for spheroid and traditional, please fix or clarify

In the converted word document, the figure lost some of its detail. The uploaded .tiff should have adjusted colors for the plot and legend.

3. Journal Edits

3.1. Please ensure that your manuscript meets PLOS ONE's style requirements, including those for file naming. The PLOS ONE style templates can be found at 

The document has been reformatted accordingly.

3.2. We note that the grant information you provided in the ‘Funding Information’ and ‘Financial Disclosure’ sections do not match. When you resubmit, please ensure that you provide the correct grant numbers for the awards you received for your study in the ‘Funding Information’ section.

These sections have been updated to be consistent.

3.3. Thank you for stating the following in the Acknowledgments Section of your manuscript: 

This work used the Extreme Science and Engineering Discovery Environment (XSEDE), which is supported by National Science Foundation grant number ACI-1548562. Anvil at Purdue and Expanse at UCSD were used through allocation TG-MDE220002. This research was done using services provided by the OSG Consortium (87–89), which is supported by the National Science Foundation awards #2030508 and #1836650. We also thank Lev Gorenstein and the rest of the Research Computing Staff for their assistance with batch computing at the Rosen Center for Advanced Computing. We would also like to acknowledge Catherine Weathered for her mentorship and her work setting up the simulation foundations in Repast and Slurm.

Remove funding information from acknowledgments and moved to funding statement.

Updated acknowledgements

We thank Lev Gorenstein and the rest of the Research Computing Staff for their assistance with batch computing at the Rosen Center for Advanced Computing. We would also like to acknowledge Catherine Weathered for her mentorship and her work setting up the simulation foundations in Repast and Slurm.

New funding statement

This work used the Extreme Science and Engineering Discovery Environment (XSEDE), which is supported by National Science Foundation grant number ACI-1548562. Anvil at Purdue and Expanse at UCSD were used through allocation TG-MDE220002. This research was done using services provided by the OSG Consortium (87–90), which is supported by the National Science Foundation awards #2030508 and #1836650. The work of NdP is made possible through funding by the National Institute of Allergy and Infectious Diseases (NIAID), NIH, through contract #75N93019C0070. The work of AH is made possible in part by Grant TL1TR002531 (T. Hurley, PI) from the National Institutes of Health, National Center for Advancing Translational Sciences, Clinical and Translational Sciences Award.

3.4. Please remove your figures from within your manuscript file, leaving only the individual TIFF/EPS image files, uploaded separately.

They are now uploaded separately and have been deleted from the manuscript.

---

## [Decision Letter · Decision Letter 1]

23 Jan 2024

PONE-D-23-27966R1In silico agent-based modeling approach to characterize multiple in vitro tuberculosis infection modelsPLOS ONE

Dear Dr. Pienaar,

Thank you for submitting your manuscript to PLOS ONE. After careful consideration, we feel that it has merit but does not fully meet PLOS ONE’s publication criteria as it currently stands. Therefore, we invite you to submit a revised version of the manuscript that addresses the points raised during the review process.

Please consider the comments from reviewer #2 and update the GitHub repository accordingly.

We look forward to receiving your revised manuscript.

Kind regards,

Bhanwar Lal Puniya, Ph.D.

Academic Editor

PLOS ONE

Journal Requirements:

Reviewers' comments:

Reviewer's Responses to Questions

**Comments to the Author**

1. If the authors have adequately addressed your comments raised in a previous round of review and you feel that this manuscript is now acceptable for publication, you may indicate that here to bypass the “Comments to the Author” section, enter your conflict of interest statement in the “Confidential to Editor” section, and submit your "Accept" recommendation.

Reviewer #1: All comments have been addressed

Reviewer #2: All comments have been addressed

2. Is the manuscript technically sound, and do the data support the conclusions?

Reviewer #1: Yes

Reviewer #2: Yes

3. Has the statistical analysis been performed appropriately and rigorously? 

Reviewer #1: Yes

Reviewer #2: Yes

4. Have the authors made all data underlying the findings in their manuscript fully available?

Reviewer #1: Yes

Reviewer #2: Yes

5. Is the manuscript presented in an intelligible fashion and written in standard English?

Reviewer #1: Yes

Reviewer #2: Yes

6. Review Comments to the Author

Reviewer #1: (No Response)

Reviewer #2: Authors have provided a link to the GitHub repository however the ReadMe files are not updated with instructions on how to run the code, and recreate the figures. Authors are recommended to document the ReadMe file in the GitHub repository and include instructions to help re-create the figures.

7. PLOS authors have the option to publish the peer review history of their article (what does this mean?). If published, this will include your full peer review and any attached files.

Reviewer #1: No

Reviewer #2: **Yes: **Meghna Verma

---

## [Author Response · Author response to Decision Letter 1]

2 Feb 2024

“Authors have provided a link to the GitHub repository however the ReadMe files are not updated with instructions on how to run the code, and recreate the figures. Authors are recommended to document the ReadMe file in the GitHub repository and include instructions to help re-create the figures.”

Thanks for this suggestion. The GitHub holds the code to run the model. We have updated this repository with a README.md file with instructions on how to setup the code, run the model on a HPC, and process output. This new ReadMe file has been changed in release v1.0.1, and version number has been changed in the manuscript and supplementary information.

The code and data used for the figures can be found in the Zenodo. Zenodo was used instead of GitHub to allow the data and code to be uploaded together, as some of the data files exceeded the file size limits of GitHub. There is already a ReadMe.md file included in the Zenodo, which has a description of the code, instructions on how to run the code, required dependencies, and what code and data is used for each figure.

---

## [Editor Report · Decision Letter 2]

6 Feb 2024

In silico agent-based modeling approach to characterize multiple in vitro tuberculosis infection models

PONE-D-23-27966R2

Dear Dr. Pienaar,

We’re pleased to inform you that your manuscript has been judged scientifically suitable for publication and will be formally accepted for publication once it meets all outstanding technical requirements.

Kind regards,

Bhanwar Lal Puniya, Ph.D.

Academic Editor

PLOS ONE
---

## [Editor Report · Acceptance letter]

12 Mar 2024

PONE-D-23-27966R2 

PLOS ONE

Dear Dr. Pienaar, 

I'm pleased to inform you that your manuscript has been deemed suitable for publication in PLOS ONE. Congratulations! Your manuscript is now being handed over to our production team.

Kind regards, 

on behalf of

Dr. Bhanwar Lal Puniya 

Academic Editor

PLOS ONE